# LOOK BACK WHEN SURPRISED: STABILIZING REVERSE EXPERIENCE REPLAY FOR NEURAL APPROXIMATION

## ABSTRACT

Experience replay-based sampling techniques are essential to several reinforcement learning (RL) algorithms since they aid in convergence by breaking spurious correlations. The most popular techniques, such as uniform experience replay (UER) and prioritized experience replay (PER), seem to suffer from sub-optimal convergence and significant bias error, respectively. To alleviate this, we introduce a new experience replay method for reinforcement learning, called Introspective Experience Replay (IER). IER picks batches corresponding to data points consecutively before the 'surprising' points. Our proposed approach is based on the theoretically rigorous reverse experience replay (RER), which can be shown to remove bias in the linear approximation setting but can be sub-optimal with neural approximation. We show empirically that IER is stable with neural function approximation and has a superior performance compared to the state-of-the-art techniques like uniform experience replay (UER), prioritized experience replay (PER), and hindsight experience replay (HER) on the majority of tasks.

## 1 INTRODUCTION

Reinforcement learning (RL) involves learning with dependent data, and algorithms designed for independent data might behave poorly coupled with the Markovian trajectories encountered in this setting. Experience replay (Lin, 1992) involves storing the received data points in a large buffer and producing a random sample from this buffer whenever the learning algorithm requires it. Therefore experience replay is usually deployed with popular algorithms like DQN, DDPG and TD3 to achieve state-of-the-art performance (Mnih et al., 2015; Lillicrap et al., 2015). It has been shown experimentally (Mnih et al., 2015) and theoretically (Nagaraj et al., 2020) that these learning algorithms for Markovian data behave sub-optimally without experience replay. Note that we use the term "sub-optimal" when consistently observing a sub-par performance compared to the oracle. In contrast, the term "instability" refers to the setting where there is a high variance in our experiments, where only a few seeds work well. We maintain this distinction throughout our paper.

The simplest and most widely used experience replay method is the uniform experience replay (UER), where the data points stored in the buffer are sampled uniformly at random every time a data point is queried (Mnih et al., 2015). However, UER might pick uninformative data points most of the time, which may slow down the convergence. For this reason, optimistic experience replay (OER) and prioritized experience replay (PER) (Schaul et al., 2015) were introduced, where samples with higher TD error (i.e., 'surprise') are sampled more often from the buffer. Optimistic experience replay (originally called "greedy TD-error prioritisation") was shown to have a high bias, and Prioritized experience replay was proposed to solve this issue (Schaul et al., 2015). However, as shown in our experiments outside of the Atari environments, PER still suffers from the problem of high bias. Although this speeds up the learning process in many cases, there can be significant biases due to picking and choosing only specific data points, which can make this method sub-optimal. The design of experience replay continues to be an active field of research. Several other experience replay techniques like Hindsight experience replay (HER) (Andrychowicz et al., 2017), Reverse Experience Replay (RER) (Rotinov, 2019), and Topological Experience Replay (TER) (Hong et al., 2022) have been proposed. An overview of these methods in discussed in Section 2.

Even though these methods are widely deployed in practice, theoretical analyses have been very limited. Recent results on learning dynamical systems (Kowshik et al., 2021b;a) showed rigorously in a theoretical setting that RER is the conceptually-grounded algorithm when learning from Markovian data. Furthermore, this work was extended to the RL setting in Agarwal et al. (2021) to achieve efficient Q learning with linear function approximation. The RER technique achieves good performance since reverse order sampling of the data points prevents the build-up of spurious correlations in the learning algorithm. In this paper, we build on this line of work and introduce **Introspective Experience Replay** (IER). Roughly speaking, IER first picks top $k$ 'pivot' points from a large buffer according to their TD error. It then returns batches of data formed by selecting the consecutive points *temporally* before these pivot points. In essence, the algorithm *looks back when surprised*. The intuition behind our approach is linked to the fact that the agent should always associate outcomes to its past actions, just like in RER. The summary of our approach is shown in Figure 1. This technique is an amalgamation of Reverse Experience Replay (RER), and Optimistic Experience Replay (OER), which only picks the points with the highest TD error.

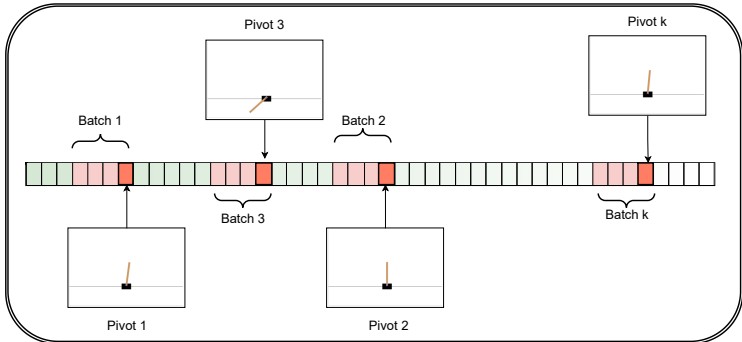

Figure 1: An illustration of our proposed methodology when selecting $k$ batches in the CartPole environment. The red color is used to indicate the batches being sampled from the replay buffer. The green samples are the un-sampled states from the buffer. The pivots are explicitly pointed by the arrow and the snapshot of the surprising state encountered.

Our main findings are summarized below:

**Better Performance Against SOTA:** Our proposed methodology (IER) outperforms previous state-of-the-art baselines such as PER, UER and HER on most environments (see Table 1, Section 5).

**Conceptual Understanding:** We consider a simple toy example where we understand the differences between UER, RER, OER and IER (Section 3.2). This study illustrates the better performance of our proposed method by showing a) why naive importance sampling, like in OER incurs a significant bias and b) why techniques like UER and RER are slow to learn.

**Forward vs. Reverse:** We show that the temporal direction (forward/reverse) to consider after picking the pivot is non-trivial. We show empirically that IER performs much better than its forward counterpart, IER (F) (see Table 3). This gives evidence towards causality playing a role in the success of IER.

**Whole vs. Component Parts:** Our method (IER) is obtained by combining two methods RER (which picks the samples in the reverse order as received) and OER (which greedily picks the samples with the largest TD error). We show that neither of these components performs well compared to their amalgamation, IER (Table 4).

**Minimal hyperparameter tuning:** Our proposed methodology uses minimal hyperparameter tuning. We use the same policy network architecture, learning rate, batch size, and all other parameters across all our runs for a given environment. These hyperparameters are selected based on the setting required to achieve SOTA performance on UER. However, we have few options available, a *Hindsight* flag, leading to the H-IER sampler, and the *Uniform Mixing fraction* leading to the U-IER sampler. Furthermore, for most of our experiments, we use a mixing fraction of 0.

Table 1: **IER outperforms previous state-of-the-art baselines**. These baselines include samplers such as UER (Mnih et al., 2013), PER (Schaul et al., 2015), and HER (Andrychowicz et al., 2017) across many environments. Results are from 13 different environments that cover a broad category of MDPs. These include a few Atari environments (previously used to support the efficacy of UER, PER), some Robotics environments (previously used to support the efficacy of HER), and many other classes of environments, including Classic Control, Box 2D, and Mujoco. More details about these experiments and their setup have been discussed in Section 5.

| Experience Replay Method | UER | PER | HER | IER |
|---|---|---|---|---|
| Best Performance Frequency | 1 | 0 | 1 | 11 |

## 2 RELATED WORKS AND COMPARISON

### 2.1 EXPERIENCE REPLAY TECHNIQUES

Experience replay involves storing consecutive temporally dependent data in a (large) buffer in a FIFO order. Whenever a learning algorithm queries for batched data, the experience replay algorithm returns a sub-sample from this buffer such that this data does not hinder the learning algorithms due to spurious correlations. The most basic form of experience replay is UER (Lin, 1992) which samples the data in the replay buffer uniformly at random. This approach has significantly improved the performance of off-policy RL algorithms like DQN (Mnih et al., 2015). Several other methods of sampling from the buffer have been proposed since; PER (Schaul et al., 2015) samples experiences from a probability distribution which assigns higher probability to experiences with significant TD error and is shown to boost the convergence speed of the algorithm. This out-performs UER in most Atari environments. HER (Andrychowicz et al., 2017) works in the "what if" scenario, where even a sub-optimal policy can lead the agent to learn what not to do and nudge the agent towards the correct action. There have also been other approaches such as Liu et al. (2019); Fang et al. (2018; 2019) have adapted HER in order to improve the overall performance with varying intuition. RER processes the data obtained in a buffer in the reverse temporal order. We refer to the following sub-section for a detailed review of this and related techniques. We will also consider 'optimistic experience replay' (OER), the naive version of PER, where at each step, only top $B$ elements in the buffer are returned when batched data is queried. This approach is known to suffer from high bias error mitigated by a sophisticated sampling procedure employed in PER. Other works such as Fujimoto et al. (2020); Pan et al. (2022) attempt to study PER and address some of its shortcomings and Lahire et al. (2021) introduces 'large batch experience replay' (LaBER) which reduces the stochastic noise in gradients, while keeping them unbiased.

### 2.2 REVERSE SWEEP TECHNIQUES

Reverse sweep or backward value iteration refers to methods that process the data as received in reverse order. This has been studied in the context of planning tabular MDPs (Dai & Hansen, 2007; Grześ & Hoey, 2013). We refer to Section 4 for a brief overview of why these methods are considered. However, this line of work assumes that the MDP and the transition functions are known. Inspired by the behavior of biological networks, Rotinov (2019) proposed reverse experience replay where the experience replay buffer is replayed in a LIFO order. Since RER forms mini-batches with consecutive data points, it is unstable with Neural approximation (i.e, it does not learn consistently across different environments). Therefore, the iterations are stabilized by 'mixing' RER with UER. However, the experiments are limited and do not demonstrate that this method outperforms even UER. A similar procedure called Episodic Backward Update (EBU) is introduced in Lee et al. (2019). However, to stabilize the pure RER , the EBU method seeks to also change the target for Q learning instead of just changing the sampling scheme in the replay buffer. The reverse sweep was independently rediscovered as RER in the context of streaming linear system identification in Kowshik et al. (2021b), where SGD with reverse experience replay was shown to achieve near-optimal performance. In contrast, naive SGD was significantly sub-optimal due to bias caused by Markovian data. The follow-up work Agarwal et al. (2021) analyzed off-policy Q learning with linear function approximation and reverse experience replay to provide near-optimal convergence guarantees using the unique super martingale structure endowed by reverse experience replay. Hong et al. (2022) considers topological experience replay, which executes reverse replay over a directed graph of observed transitions. Mixed with PER enables non-trivial learning in some challenging environments. Another line of work

(Florensa et al., 2017; Moore & Atkeson, 1993; Goyal et al., 2018; Schroecker et al., 2019) considers reverse sweep with access to a simulator or using a fitted generative model. Our work on the other hand, only seeks on-policy access to the MDP.

## 3 BACKGROUND AND PROPOSED METHODOLOGY

We consider episodic reinforcement learning (Sutton & Barto, 2018), where at each time step an agent takes actions $a_t$ in an uncertain environment with state $s_t$, and receives a reward $r_t$. The environment then evolves into a new state $s_{t+1}$ whose law depends only on $s_t, a_t$. Our goal is to (approximately) find the policy $\pi^*$ which maps the environmental state $s$ to an action $a$ such that when the agent takes the action $a_t = \pi^*(s_t)$, the discounted reward $\mathbb{E}\left[\sum_{t=0}^{\infty} \gamma^t r_t\right]$ is maximized. To achieve this, we consider algorithms like DQN (Mnih et al. (2015)), DDPG (Lillicrap et al. (2015)) and TD3 (Fujimoto et al. (2018)), which routinely use experience replay buffers. In this paper, we introduce a new experience replay method, IER, and investigate the performance of the aforementioned RL algorithms with this modification. In this work, when we say "return", we mean discounted episodic reward.

### 3.1 METHODOLOGY

We now describe our main method in a general way where we assume that we have access to a data collection mechanism $\mathbb{T}$ which samples new data points. This then appends the sampled data points to a buffer $\mathcal{H}$ and discards some older data points. The goal is to run an iterative learning algorithm $\mathbb{A}$, which learns from batched data of batch size $B$ in every iteration. We also consider an important metric $I$ associated with the problem. At each step, the data collection mechanism $\mathbb{T}$ collects a new episode and appends it to the buffer $\mathcal{H}$ and discards some old data points, giving us the new buffer as $\mathcal{H} \leftarrow \mathbb{T}(\mathcal{H})$. We then sort the entries of $\mathcal{H}$ based on the importance metric $I$ and store the indices of the top $G$ data points in an array $P = [P[0], \dots, P[G-1]]$. Then for every index in $P$, we run the learning algorithm $\mathbb{A}$ with the batch $D = (\mathcal{H}(P[i]), \dots, \mathcal{H}(P[i] - B + 1))$. In some cases, we can 'mix'[1] this with the standard UER sampling mechanism to reduce bias, as shown below. This amalgamation help mitigate bias that could arise from our important metric sampling curriculum. We describe this procedure in Algorithm 1.

In the reinforcement learning setting, $\mathbb{T}$ runs an environment episode with the current policy and appends the transitions and corresponding rewards to the buffer $\mathcal{H}$ in the FIFO order, maintaining a total of $1E6$ data points, usually. We choose $\mathbb{A}$ to be an RL algorithm like TD3 or DQN or DDPG. The importance function $I$ is the magnitude of the TD error with respect to the current Q-value estimate provided by the algorithm $\mathbb{A}$ (i.e., $I = |Q(s, a) - R(s, a) - \gamma \sup_{a'} Q^{\text{target}}(s', a')|$). When the data collection mechanism ($\mathbb{T}$) is the same as in UER, we will call this method **IER**. In optimistic experience replay (OER), we take $\mathbb{T}$ to be the same as in UER. However, we query top $BG$ data points from the buffer $\mathcal{H}$ and return $G$ disjoint batches each of size $B$ from these 'important' points. It is clear that IER is a combination of OER and RER. Notice that we can also consider the data collection mechanism like that of HER, where examples are labeled with different goals, i.e. $\mathbb{T}$ has now been made different, keeping the sampling process exactly same as before. In this case, we will call our algorithm **H-IER**. Our experiment in Enduro and Acrobat depicts an example of this successful coalition. We also consider the RER method, which served as a motivation for our proposed approach. Under this sampling methodology, the batches are drawn from $\mathcal{H}$ in the temporally reverse direction. This approach is explored in the works mentioned in Section 2.2. We discuss this methodology in more detail in Appendix F.

### 3.2 DIDACTIC TOY EXAMPLE

In this section, we discuss the working of IER on a simple environment such as GridWorld-1D, and compare this with some of our baselines such as UER, OER, RER, and IER (F). In this environment, the agent lives on a discrete 1-dimensional grid of size 40 with a max-timestep of 1000 steps, and at each time step, the agent can either move left or right by one step. The agent starts from the *starting state* (S; [6]), the goal of the agent is to reach *goal state* (G; [40]) getting a reward of $+1$, and there

---

[1]Mixing here denotes sampling with a given probability from one sampler A, and filling the remaining samples of a batch with sampler B.

---

**Algorithm 1:** Our proposed Introspective Experience Replay (IER) for Reinforcement Learning

---

**Input:** Data collection mechanism $\mathbb{T}$, Data buffer $\mathcal{H}$, Batch size $B$, grad steps per Epoch $G$,
number of episodes $N$, Importance function $I$, learning procedure $\mathbb{A}$, Uniform Mixing
fraction $p$

$n \leftarrow 0$;
**while** $n < N$ **do**
    $n \leftarrow n + 1$;
    $\mathcal{H} \leftarrow \mathbb{T}(\mathcal{H})$                                   ▷ Add a new episode to the buffer
    $\mathcal{I} \leftarrow I(\mathcal{H})$                 ▷ Compute importance of each data point in the buffer
    $P \leftarrow \mathsf{Top}(\mathcal{I}; G)$                         ▷ Obtain index of top $G$ elements of $\mathcal{I}$
    $g \leftarrow 0$;
    **while** $g < G$ **do**
        **if** $g < (1-p)G$ **then**
            $D \leftarrow \mathcal{H}[P[g] - B, P[g]]$       ▷ Load batch of previous $B$ examples from pivot $P[g]$
        **else**
            $D \leftarrow \mathcal{H}[\mathsf{Uniform}(\mathcal{H}, B)]$       ▷ Randomly chose $B$ indices from buffer
        **end**
        $g \leftarrow g + 1$;
        $\mathbb{A}(D)$                     ▷ Run the learning algorithm with batch data $D$
    **end**
**end**

---

is also a *trap state* (T; [3]), where the agents gets a reward of $-2$. The reward in every other state is $0$. For simplicity, we execute an offline exploratory policy where the agent moves left or right with a probability of half and obtain a buffer of size 30000. The rewarding states occur very rarely in the buffer since it is hard to reach for this exploration policy. The episode ends upon meeting either of two conditions: (i) the agent reaches the terminal state, which is the *goal state*, or (ii) the agent has exhausted the max-timestep condition and has not succeeded in reaching any terminal state. An overview of our toy environment is depicted in Figure 2(a). Other hyperparameters crucial to replicating this experiment are described in Appendix B.

In this example, reaching the goal state as quickly as possible is vital to receive a positive reward and avoid the fail state. Therefore, it is essential to understand the paths which reach the goal state. Figure 2(b) depicts the number of times each state occurs in the buffer. Furthermore, the remaining subplots of Figure 2 depict the *Absolute Frequency* of our off-policy algorithm trained in this environment. A state's "absolute frequency" is the number of times the replay technique samples a given state during the algorithm's run. The experiments on this simple didactic toy environment do highlight a few interesting properties:

**Comparison of UER and IER:** Since the goal state appears very rarely in buffer, UER and RER rarely sample the goal state and hence do not manage to learn effectively. While RER naturally propagates the information about the reward back in time to the states that led to the reward, it does not often sample the rewarding state.

**Limitation of OER:** While OER samples a lot from the states close to the goal state, the information about the reward does not propagate to the start state. We refer to the bottleneck in Figure 2(e) where some intermediate states are not sampled.

**Advantage of IER:** IER prioritizes sampling from the goal state and propagates the reward backward so that the entire path leading to the reward is now aware of how to reach the reward. Therefore, a combination of RER and OER reduces the sampling bias in OER by preventing the bottlenecks seen in Figure 2(e).

**Bottleneck of IER (F):** IER (F) has a more significant bottleneck when compared to RER and chooses to sample the non-rewarding middle states most often. Also, note that whenever IER (F) chooses the goal state as the pivot, it selects the rest of the batch to overflow into the next episode, which begins at the starting state. This does not allow the algorithm to effectively learn the path which *led* to the goal state.

The toy example above models several salient features in more complicated RL environments.

(i) In the initial stages of learning, the exploratory policy is essentially random, and such a naive exploratory policy does not often lead to non-trivial rewards.

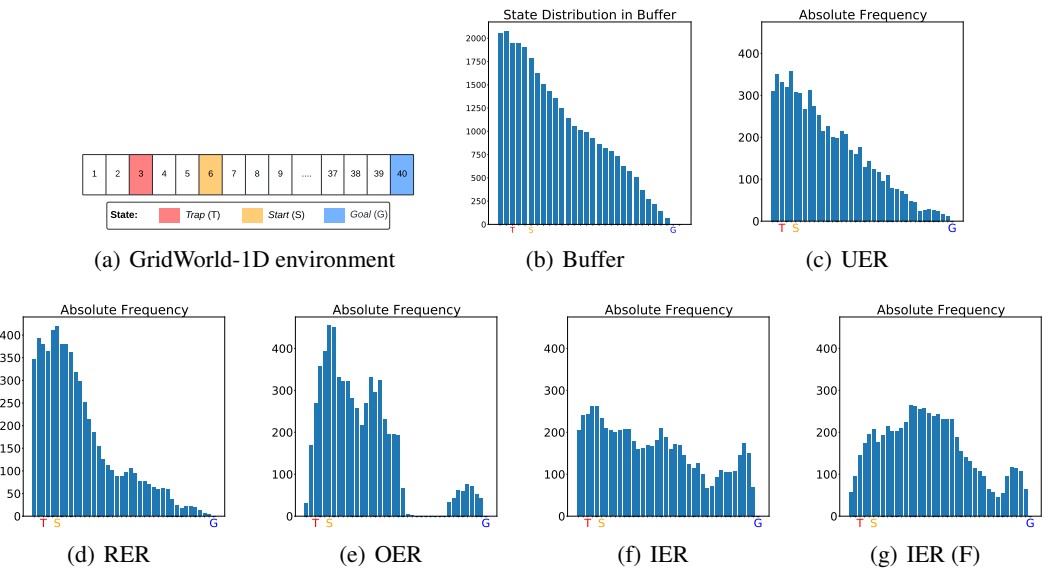

Figure 2: Gridworld-1D environment is depicted in Figure 2(a). Distribution of states in the buffer (Figure 2(b)) and relative frequency of different experience replay samplers on the didactic example of GridWorld-1D environment (Figure 2(c);2(d);2(e);2(f);2(g)).

(ii) Large positive and negative reward states (the goal and trap states), and their neighbors provide the pivot state for IER and OER.

We show empirically that this holds in more complicated environments as well. Figure 3 depicts the surprise vs. reward for the Ant environment. Here we see a strong correlation between absolute reward and TD error ("Surprise factor").

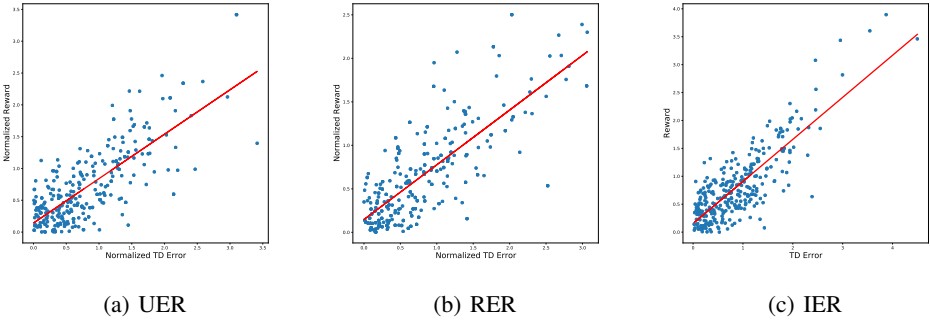

Figure 3: Relationship between absolute values of TD Error (Surprise factor) and Reward for the Ant environment.

## 4    UNDERSTANDING REVERSE REPLAY

There are various conceptual ways we can look at RER and IER. This section outlines some of the motivations behind using this technique. Theoretical works such as Agarwal et al. (2021); Kowshik et al. (2021b) have established rigorous theoretical guarantees for RER by utilizing super martingale structures. This structure is not present in forward replay techniques (i.e., the opposite of reverse replay) as shown in Kowshik et al. (2021a). We refer to Appendix D where we show via an ablation study that going forward in time instead of reverse does not work very well. We give the following explanations for the success of IER.

**Propogation of Sparse Rewards:** In many RL problems, non-trivial rewards are sparse and only received at the goal states. Therefore, processing the data backward in time from such goal states helps the algorithm learn about the states that led to this non-trivial reward. Our study (see Figure 3 and Appendix E for further details) shows that in many environments IER picks pivots which are the states with large (positive or negative) rewards, enabling effective learning.

**Bias Reduction in OER:** OER, which greedily chooses the examples with the largest TD error to learn from, performs very poorly due to bias. To illustrate this phenomenon, we refer to the didactic example in Section 3.2. One possible way of viewing IER is that RER is used to reduce the bias in OER. Indeed, the theoretical analysis in Agarwal et al. (2021) shows that RER removes bias in Q-learning type algorithms, albeit in a different sense. PER achieves a similar bias reduction for OER with a sophisticated and expensive sampling scheme over the buffer.

**Causality:** MDPs have a natural causal structure: actions and events in the past influence the events in the future. Therefore, whenever we see a surprising or an unexpected event, we can understand why or how it happened by looking into the past. Further theoretical work is needed to realize IER better.

We further illustrate the same by referring to the straightforward didactic example (Section 3.2) where we can see the effects of each of the experience replay methods. We also demonstrate superior performance on more complicated environments (Section 5) showcasing the robustness of our approach with minimal hyperparameter tuning.

## 5 EXPERIMENTAL RESULTS

In this section, we briefly discuss our experimental setup as well as the results from our experiments.

**Environments:** We evaluate our approach on a diverse class of environments, such as (i) Environments with low-dimensional state space (including classic control and Box-2D environments), (ii) Multiple joint dynamic simulation and robotics environments (including Mujoco and Robotics environments), and (iii) Human-challenging environments (such as Atari environments). Note that previous seminal papers in the field of experience replay such as Mnih et al. (2013), Schaul et al. (2015), and Andrychowicz et al. (2017) showed the efficacy of their approach on a subset of these classes. For instance, UERand PER was shown to work well on Atari games. Furthermore, HER was effective in the Robotics environments such as FetchReach. In this work, we perform a more extensive study to show the robustness and effectiveness of our model not just in Atari and Robotics environments but also in Mujoco, Box2D, and Classic Control environments. Due to computational limitations and the non-reproducibility of some baselines, we could not extend our experiments to some Atari environments. We refer to Appendix A for a brief description of the environments used.

**Hyperparameters:** Refer to Appendix B for the exact hyperparameters used. Across all our experiments on various environments, we use a standard setting for all the different experience replay buffers. This classic setting is set so we can reproduce state-of-the-art performance using UER on the respective environment. For most of our experiments, we set the uniform mixing fraction ($p$) from Algorithm 1 to be 0. We use a non-zero $p$ value only for a few environments to overcome the bias while training, as described in Appendix B. For PER, we use the same hyperparameters used in the Schaul et al. (2015) paper which had been proven robust across 50 different environments of Atari.

**Metric:** To compare the different models, we use the *top-k seeds moving average return* as the evaluation metric across all our runs. Top-k seeds here mean we take the average of $k = 3$ seeds that gave the best performance. It is common to use top-k trials to be selected from among many trials in the reinforcement learning literature (see Schaul et al. (2015);Schaff et al. (2019);Sarmad et al. (2019);Wu et al. (2017);Mnih et al. (2016)). This factors in the seed sensitivity. Moving average with a given window size is taken for learning curves (with a window size of 20 for FetchReach and 50 for all others) to reduce the variation in return which is inherently present in each epoch. We argue that taking a moving average is essential since, usually, pure noise can be leveraged to pick a time instant where a given method performs best (Henderson et al., 2018). Considering the top-k seed averaging of the last step performance of the moving average of the learning curves gives our metric - the *top-k seeds moving average return*.

**Comparison with SOTA:** Table 2 depicts our results in various environments upon using different SOTA replay sampler mechanisms (UER, PER and HER). Our proposed sampler outperforms all other baselines in most tasks and compares favorably in others. Our experiments on various environments across various classes, such as classic control, Atari, etc., show that our proposed methodology consistently outperforms all other baselines in most environments, as summarized in Table 1. Furthermore, our proposed methodology is robust across various environments, as highlighted in Table 2. The learning curves for our experiments have been depicted in the Appendix. (see Appendix C)

Table 2: *Top-k seeds Moving Average Return* results across various environments. From our experiments, we note that IER outperforms previous SOTA baselines in most environments. Appendix B depicts the hyperparameters used for the experiment.

| Dataset | *UER* | *PER* | *HER* | *IER* |
|---|---|---|---|---|
| CartPole | $153.14_{\pm 32.82}$ | $198.06_{\pm 3.68}$ | $173.84_{\pm 26.11}$ | $\mathbf{199.83}_{\pm 0.31}$ |
| Acrobot | $-257.93_{\pm 184.28}$ | $-291.56_{\pm 148.84}$ | $-389.42_{\pm 113.22}$ | $\mathbf{-193.90}_{\pm 57.56}$ |
| Inverted Pendulum | $-161.93_{\pm 10.55}$ | $-171.73_{\pm 10.55}$ | $-629.42_{\pm 815.24}$ | $\mathbf{-150.27}_{\pm 9.63}$ |
| LunarLander | $-4.42_{\pm 20.06}$ | $5.33_{\pm 16.10}$ | $6.00_{\pm 10.02}$ | $\mathbf{12.32}_{\pm 27.55}$ |
| HalfCheetah | $10808.93_{\pm 1094.32}$ | $99.75_{\pm 1124.46}$ | $\mathbf{11072.54}_{\pm 297.12}$ | $10544.88_{\pm 342.01}$ |
| Ant | $3932.85_{\pm 1024.86}$ | $-2699.84_{\pm 1.34}$ | $3803.17_{\pm 996.81}$ | $\mathbf{4203.21}_{\pm 345.22}$ |
| Reacher | $\mathbf{-4.97}_{\pm 0.31}$ | $-5.42_{\pm 0.61}$ | $-5.30_{\pm 0.50}$ | $-4.92_{\pm 0.27}$ |
| Walker | $3597.03_{\pm 1203.79}$ | $1709.48_{\pm 1635.90}$ | $889.82_{\pm 1427.91}$ | $\mathbf{4349.29}_{\pm 680.35}$ |
| Hopper | $3072.65_{\pm 621.40}$ | $0.49_{\pm 2.83}$ | $2685.72_{\pm 1221.81}$ | $\mathbf{3205.05}_{\pm 406.35}$ |
| Inverted Double Pendulum | $8489.54_{\pm 927.69}$ | $7163.77_{\pm 3404.14}$ | $9002.05_{\pm 464.20}$ | $\mathbf{9067.69}_{\pm 402.39}$ |
| Fetch-Reach | $-1.84_{\pm 0.57}$ | $-49.90_{\pm 0.10}$ | $-2.92_{\pm 1.79}$ | $\mathbf{-1.74}_{\pm 0.24}$ |
| Pong | $\mathbf{19.15}_{\pm 1.32}$ | $17.02_{\pm 3.27}$ | $18.70_{\pm 1.02}$ | $19.10_{\pm 1.20}$ |
| Enduro | $227.01_{\pm 319.15}$ | $565.23_{\pm 116.36}$ | $514.32_{\pm 132.39}$ | $\mathbf{586.32}_{\pm 111.44}$ |

**Forward vs. Reverse:** The intuitive limitation to the "looking forward" approach is that in many RL problems, the objective for the agent is to reach a final goal state, where the non-trivial reward is obtained. Since non-trivial rewards are only offered in this goal state, it is informative to look back from here to learn about the states that *lead* to this. When the goals are sparse, the TD error is more likely to be large upon reaching the goal state. Our algorithm selects these as pivots, and IER (F) might select batches overflowing into the next episode. Our studies on many environments (see Figure 3 and Appendix E) show that the pivot points selected based on importance indeed have large (positive or negative) rewards. Our experiments depicted in Table 3 show that IER outperforms IER (F) in most environments.

Table 3: *Top-k seeds Moving Average Return* results across various environments for Temporal Ablation study between IER (F) and IER (R; default). Note that we use the base setting of IER in this section to avoid spurious comparisons (i.e, with $p = 0$ and no hindsight).

| Dataset | *Forward* | *Reverse* |
|---|---|---|
| CartPole | $196.51_{\pm 6.26}$ | $\mathbf{199.83}_{\pm 0.31}$ |
| Acrobot | $-423.15_{\pm 108.08}$ | $\mathbf{-313.03}_{\pm 190.27}$ |
| Inverted Pendulum | $\mathbf{-882.13}_{\pm 521.77}$ | $-1111.51_{\pm 559.83}$ |
| LunarLander | $-22.75_{\pm 30.19}$ | $\mathbf{12.32}_{\pm 27.55}$ |
| HalfCheetah | $9369.30_{\pm 454.40}$ | $\mathbf{10108.15}_{\pm 919.27}$ |
| Ant | $2963.71_{\pm 828.50}$ | $\mathbf{4203.21}_{\pm 345.22}$ |
| Reacher | $-5.25_{\pm 0.33}$ | $\mathbf{-4.92}_{\pm 0.27}$ |
| Walker | $2213.89_{\pm 1684.03}$ | $\mathbf{2830.03}_{\pm 881.51}$ |
| Hopper | $393.64_{\pm 181.89}$ | $\mathbf{505.58}_{\pm 266.52}$ |
| Inverted Double Pendulum | $\mathbf{9260.27}_{\pm 95.59}$ | $9067.69_{\pm 402.39}$ |
| FetchReach | $-17.73_{\pm 27.91}$ | $\mathbf{-2.28}_{\pm 1.11}$ |
| Pong | $17.92_{\pm 2.60}$ | $\mathbf{19.10}_{\pm 1.20}$ |
| Enduro | $\mathbf{600.87}_{\pm 149.99}$ | $525.84_{\pm 146.39}$ |

**Whole vs. Component Parts:** Our approach is an amalgamation of OER and RER. Here we compare these individual parts with IER. Table 4 describes the *Top-k seeds Moving Average Return* across various environments in this domain. As demonstrated, IER outperforms its component parts OER nor RER .

Table 4: *Top-k seeds Moving Average Return* results across various environments for ablation study between RER, OER, and IER.

| Dataset | RER | OER | IER |
|---|---|---|---|
| CartPole | 163.93 ± 40.03 | 162.36 ± 34.89 | **199.83** ± 0.31 |
| Acrobot | -320.70 ± 144.05 | -472.63 ± 31.21 | **-193.90** ± 57.56 |
| Inverted Pendulum | -735.38 ± 613.5 | -166.57 ± 17.23 | **-150.27** ± 9.63 |
| LunarLander | 9.84 ± 13.23 | -16.08 ± 15.38 | **12.32** ± 27.55 |
| HalfCheetah | 9449.39 ± 648.60 | 2237.91 ± 2824.72 | **10544.88** ± 342.01 |
| Ant | 2168.47 ± 415.53 | -47.93 ± 20.47 | **4203.21** ± 345.22 |
| Reacher | -5.91 ± 0.37 | -5.28 ± 0.58 | **-4.92** ± 0.27 |
| Walker | 1578.33 ± 1313.11 | 207.51 ± 193.06 | **4349.29** ± 680.35 |
| Hopper | 206.23 ± 318.49 | 660.24 ± 580.77 | **3205.05** ± 406.35 |
| Inverted Double Pendulum | 8953.44 ± 456.95 | 7724.99 ± 1726.58 | **9067.69** ± 402.39 |
| Fetch-Reach | -49.94 ± 0.07 | -47.72 ± 3.33 | **-1.74** ± 0.24 |
| Pong | 18.58 ± 1.75 | 3.52 ± 21.33 | **19.10** ± 1.20 |
| Enduro | 483.98 ± 75.45 | 361.21 ± 86.30 | **586.32** ± 111.44 |

## 6    DISCUSSION AND CONCLUSIONS

We summarize our results and discuss possible future steps.

**Speedup:**    IER shows a significant speedup in terms of time complexity over PER as depicted in Table 5. On average IER achieves a speedup improvement of $26.20\%$ over PER across a large umbrella of environment classes. As the network becomes more extensive, our approach does have a higher overhead (especially computing TD error). Future work can investigate how to further reduce the computational complexity of our method by computing the TD error fewer times at the cost of operating with an older TD error. We also notice a speedup of convergence towards an optimal policy of our proposed approach, as shown on few environments. Furthermore, the lack of speedup in some of the other experiments (even if they offer an overall performance improvement) could be due to the fact that the "surprised" pivot cannot be successfully utilized to teach the agent rapidly in the initial stages. We refer to Appendix C for the learning curves.

**Issues with stability and consistency**    Picking pivot points by looking at the TD error might cause more biases and instability compared to UER as seen in some environments like HalfCheetah, LunarLander, and Ant, where there is a sudden drop in performance for some episodes (see Appendix C). We observe that our strategy IER corrects itself quickly, unlike RER, which cannot do this (see Figure 5(e)). Increasing the number of pivot points per episode (the parameter $G$) and the uniform mixing probability $p$ usually mitigates this. However, these steps might slow the initial convergence to a locally optimal policy. In this work, we do not focus on tuning these hyper-parameters since our objective was to obtain methods that require minimal hyper-parameter tuning. However, future work can systematically investigate the significance of these parameters in various environments.

Table 5:  Average Speedup in terms of time complexity over PER across various environment classes.

| Environment | Average Speedup |
|---|---|
| Classic Control | 32.66% ↑ |
| Box-2D | 54.32% ↑ |
| Mujoco | 18.09% ↓ |
| Robotics | 55.56% ↑ |
| Atari | 6.53% ↑ |

**Why does IER outperform the traditional RER?**    The instability of pure RER with neural approximation has been noted in various works (Rotinov, 2019; Lee et al., 2019), where RER is stabilized by mixing it with UER . Hong et al. (2022) stabilizes reverse sweep by mixing it with PER. This is an interesting phenomenon since RER is near-optimal in the tabular and linear approximation settings (Agarwal et al., 2021). Two explanations of this are i) The loss function used to train the neural network is highly non-convex, which hinders the working of RER and ii) The proof given in Agarwal et al. (2021) relies extensively on 'coverage' of the entire state-action space - that is, the entire state-action space is visited enough number of times - which might not hold, as shown in the toy example in Section 3.2.

## REPRODUCIBILITY STATEMENT

In this paper, we work with thirteen datasets, all of which are open-sourced in gym (https://github.com/openai/gym). More information about the environments is available in Appendix A. We predominantly use DQN, DDPG and TD3 algorithms in our research, both of which have been adapted from their open-source code. We also experimented with seven different replay buffer methodologies, all of which have been adapted from their source code[2]. More details about the models and hyperparameters are described in Appendix B. All runs have been run using the A100-SXM4-40GB, TITAN RTX, and V100 GPUs.

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

# A   ENVIRONMENTS

For all OpenAI environments, data is summarized from https://github.com/openai/gym, and more information is provided in the wiki https://github.com/openai/gym/wiki. Below we briefly describe some of the tasks we experimented on in this paper.

## A.1   CARTPOLE-V0

CartPole, as introduced in Barto et al. (1983), is a task of balancing a pole on top of the cart. The cart has access to position and velocity as its state vector. Furthermore, it can go either left or right for each action. The task is over when the agent achieves 200 timesteps without a positive reward (balancing the pole) which is the goal state or has failed, either when (i) the cart goes out of boundaries ($\pm$ 2.4 units off the center), or (ii) the pole falls over (less than $\pm$ 12 deg). The agent is given a continuous 4-dimensional space describing the environment and can respond by returning one of two values, pushing the cart either right or left.

## A.2   ACROBOT-V1

Acrobot, as introduced in Sutton (1995), is a task where the agent is given rewards for swinging a double-jointed pendulum up from a stationary position. The agent can actuate the second joint by one of three actions: left, right, or no torque. The agent is given a six-dimensional vector comprising the environment's angles and velocities. The episode terminates when the end of the second pole is over the base. Each timestep that the agent does not reach this state gives a -1 reward, and the episode length is 500 timesteps.

## A.3   PENDULUM-V0

The inverted pendulum swingup problem, as introduced in Lillicrap et al. (2015) is based on the classic problem in control theory. The system consists of a pendulum attached fixed at one end, and free at the other end. The pendulum starts in a random position and the goal is to apply torque on the free end to swing it into an upright position, with its center of gravity right above the fixed point. The episode length is 200 timesteps, and the maximum reward possible is 0, when no torque is being applied, and the object has 0 velocity remaining at an upright configuration.

## A.4   LUNARLANDER-V2

The LunarLander environment introduced in Brockman et al. (2016) is a classic rocket trajectory optimization problem. The environment has four discrete actions - do nothing, fire the left orientation engine, fire the right orientation engine, and fire the main engine. This scenario is per Pontryagin's maximum principle, as it is optimal to fire the engine at full throttle or turn it off. The landing coordinates (goal) is always at $(0,0)$. The coordinates are the first two numbers in the state vector. There are a total of 8 features in the state vector. The episode terminates if (i) the lander crashes, (ii) the lander gets outside the window, or (iii) the lander does not move nor collide with any other body.

## A.5   HALFCHEETAH-V2

HalfCheetah is an environment based on the work by Wawrzyński (2009) adapted by Todorov et al. (2012). The HalfCheetah is a 2-dimensional robot with nine links and eight joints connecting them (including two paws). The goal is to apply torque on the joints to make the cheetah run forward (right) as fast as possible, with a positive reward allocated based on the distance moved forward and a negative reward is given for moving backward. The torso and head of the cheetah are fixed, and the torque can only be applied to the other six joints over the front and back thighs (connecting to the torso), shins (connecting to the thighs), and feet (connecting to the shins). The reward obtained by the agent is calculated as follows:

$$r_t = \dot{x_t} - 0.1 * \|a_t\|_2^2$$

## A.6 ANT-V2

Ant is an environment based on the work by Schulman et al. (2015) and adapted by Todorov et al. (2012). The ant is a 3D robot with one torso, a free rotational body, and four legs. The task is to coordinate the four legs to move in the forward direction by applying torques on the eight hinges connecting the two links of each leg and the torso. Observations consist of positional values of different body parts of the ant, followed by the velocities of those individual parts (their derivatives), with all the positions ordered before all the velocities. The reward obtained by the agent is calculated as follows:

$$r_t = \dot{x}_t - 0.5 * \|a_t\|_2^2 - 0.0005 * \left\| s_t^{\text{contact}} \right\|_2^2 + 1$$

## A.7 REACHER-V2

The Reacher environment, as introduced in Todorov et al. (2012), is a two-jointed robot arm. The goal is to move the robot's end effector (called *fingertip*) close to a target that is spawned at a random positions. The action space is a two-dimensional vector representing the torque to be applied at the two joints. The state space consists of angular positions (in terms of cosine and sine of the angle formed by the two moving arms), coordinates, and velocity states for different body parts followed by the distance from target for the whole object.

## A.8 HOPPER-V2

The Hopper environment, as introduced in Todorov et al. (2012), sets out to increase the number of independent state and control variables compared to classic control environments. The hopper is a two-dimensional figure with one leg that consists of four main body parts - the torso at the top, the thigh in the middle, the leg at the bottom, and a single foot on which the entire body rests. The goal of the environment is to make hops that move in the forward (right) direction by applying torques on the three hinges connecting the body parts. The action space is a three-dimensional element vector. The state space consists of positional values for different body parts followed by the velocity states of individual parts.

## A.9 WALKER-V2

The Walker environment, as builds on top of the Hopper environment introduced in Todorov et al. (2012), by adding another set of legs making it possible for the robot to walker forward instead of hop. The hopper is a two-dimensional figure with two legs that consists of four main body parts - the torso at the top, two thighs in the middle, two legs at the bottom, and two feet on which the entire body rests. The goal of the environment is to coordinate both feel and move in the forward (right) direction by applying torques on the six hinges connecting the body parts. The action space is a six-dimensional element vector. The state space consists of positional values for different body parts followed by the velocity states of individual parts.

## A.10 INVERTED DOUBLE-PENDULUM-V2

Inverted Double-Pendulum as introduced in Todorov et al. (2012) is built upon the CartPole environment as introduced in Barto et al. (1983), with the infusion of Mujoco. This environment involves a cart that can be moved linearly, with a pole fixed and a second pole on the other end of the first one (leaving the second pole as the only one with one free end). The cart can be pushed either left or right. The goal is to balance the second pole on top of the first pole, which is on top of the cart, by applying continuous forces on the cart. The agent takes a one-dimensional continuous action space in the range [-1,1], denoting the force applied to the cart and the sign depicting the direction of the force. The state space consists of positional values of different body parts of the pendulum system, followed by the velocities of those individual parts (their derivatives) with all the positions ordered before all the velocities. The goal is to balance the double-inverted pendulum on the cart while maximizing its height off the ground and having minimum disturbance in its velocity.

### A.11 FETCHREACH-V1

The FetchReach environment introduced in Plappert et al. (2018) was released as part of *OpenAI Gym* and used the Mujoco physics engine for fast and accurate simulation. The goal is 3-dimensional and describes the desired position of the object. Rewards in this environment are sparse and binary. The agent obtains a reward of $0$ if the target location is at the target location (within a tolerance of $5$ cm) and $-1$ otherwise. Actions are four-dimensional, where 3 specifies desired gripper movement, and the last dimension controls the opening and closing of the gripper. The FetchReach aims to move the gripper to a target position.

### A.12 PONG-V0

Pong, also introduced in Mnih et al. (2013), is comparatively more accessible than other Atari games such as Enduro. Pong is a two-dimensional sports game that simulates table tennis. The player controls an in-game paddle by moving vertically across the left and right sides of the screen. Players use this paddle to hit the ball back and forth. The goal is for each player to reach eleven points before the opponent, where the point is earned for each time the agent returns the ball and the opponent misses.

### A.13 ENDURO-V0

Enduro, introduced in Mnih et al. (2013), is a hard environment involving maneuvering a race car in the National Enduro, a long-distance endurance race. The goal of the race is to pass a certain number of cars each day. The agent must pass 200 cars on the first day and 300 cars on all subsequent days. Furthermore, as time passes, the visibility changes as well. At night in the game, the player can only see the oncoming cars' taillights. As the days' progress, cars will become more challenging to avoid. Weather and time of day are factors in how to play. During the day, the player may drive through an icy patch on the road, which would limit control of the vehicle, or a patch of fog may reduce visibility.

## B MODEL AND HYPERPARAMETERS

In this paper, we work with two classes of algorithms: DQN, DDPG and TD3. The hyperparameters used for training our DQN algorithms in various environments are described in Table 6. The hyperparameters used for training our DDPG algorithms in various environments are described in Table 7. The hyperparameters used for training DDPG are described in Table 7. Furthermore, the hyperparameters used for training TD3 are described in Table 8.

Table 6: Hyperparameters used for training DQN on various environments.

| Description | CartPole | Acrobot | LunarLander | Pong | Enduro | argument_name |
|---|---|---|---|---|---|---|
| *General Settings* | | | | | | |
| Discount | 0.9 | 0.9 | 0.9 | 0.99 | 0.99 | discount |
| Batch size | 512 | 512 | 512 | 32 | 32 | batch_size |
| Number of epochs | 100 | 100 | 200 | 150 | 800 | n_epochs |
| Steps per epochs | 10 | 10 | 10 | 20 | 20 | steps_per_epoch |
| Number of train steps | 500 | 500 | 500 | 125 | 125 | num_train_steps |
| Target update frequency | 30 | 30 | 10 | 2 | 2 | target_update_frequency |
| Replay Buffer size | $1e^6$ | $1e^6$ | $1e^6$ | $1e^4$ | $1e^4$ | buffer_size |
| *Algorithm Settings* | | | | | | |
| CNN Policy Channels | - | - | - | $(32, 64, 64)$ | $(32, 64, 64)$ | cnn_channel |
| CNN Policy Kernels | - | - | - | $(8, 4, 3)$ | $(8, 4, 3)$ | cnn_kernel |
| CNN Policy Strides | - | - | - | $(4, 2, 1)$ | $(4, 2, 1)$ | cnn_stride |
| Policy hidden sizes (MLP) | $(8, 5)$ | $(8, 5)$ | $(8, 5)$ | $(512, )$ | $(512, )$ | pol_hidden_sizes |
| Buffer batch size | 64 | 128 | 128 | 32 | 32 | batch_size |
| *Exploration Settings* | | | | | | |
| Max epsilon | 1.0 | 1.0 | 1.0 | 1.0 | 1.0 | max_epsilon |
| Min epsilon | 0.01 | 0.1 | 0.1 | 0.01 | 0.01 | min_epsilon |
| Decay ratio | 0.4 | 0.4 | 0.12 | 0.1 | 0.1 | decay_ratio |
| *Optimizer Settings* | | | | | | |
| Learning rate | $5e^{-5}$ | $5e^{-5}$ | $5e^{-5}$ | $1e^{-4}$ | $1e^{-4}$ | lr |
| *IER Specific Settings* | | | | | | |
| Use Hindsight for storing states | – | ✓ | – | – | ✓ | use_hindsight |
| Mixing Factor (p) | 0 | 0 | 0 | 0 | 0 | p |

Table 7: Hyperparameters used for training DDPG on Pendulum environment.

| Description | Pendulum | argument_name |
|---|---|---|
| *General Settings* | | |
| Discount | 0.95 | `discount` |
| Batch size | 256 | `batch_size` |
| Number of epochs | 50 | `n_epochs` |
| Steps per epochs | 50 | `steps_per_epoch` |
| Number of train steps | 40 | `num_train_steps` |
| Target update Tau | 0.01 | `target_update_frequency` |
| Replay Buffer size | $1e^6$ | `buffer_size` |
| *Algorithm Settings* | | |
| Policy hidden sizes (MLP) | $(400, 300)$ | `pol_hidden_sizes` |
| QF hidden sizes (MLP) | $(400, 300)$ | `qf_hidden_sizes` |
| Buffer batch size | 256 | `batch_size` |
| *Exploration Settings* | | |
| Exploration Policy | Ornstein Uhlenbeck Noise | `exp_policy` |
| Sigma | 0.2 | `sigma` |
| *Optimizer Settings* | | |
| Policy Learning rate | $1e^{-4}$ | `pol_lr` |
| QF Learning rate | $1e^{-3}$ | `qf_lr` |
| *IER Specific Settings* | | |
| Use Hindsight for storing states | − | `use_hindsight` |
| Mixing Factor (p) | 0 | `p` |

Table 8: Hyperparameters used for training TD3 on various environments.

| Description | Ant | Reacher | Walker | Double-Pendulum | HalfCheetah | Hopper | FetchReach | argument_name |
|---|---|---|---|---|---|---|---|---|
| *General Settings* | | | | | | | | |
| Discount | 0.99 | 0.99 | 0.99 | 0.99 | 0.99 | 0.99 | 0.95 | `discount` |
| Batch size | 250 | 250 | 250 | 100 | 100 | 100 | 256 | `batch_size` |
| Number of epochs | 500 | 500 | 500 | 750 | 500 | 500 | 100 | `n_epochs` |
| Steps per epochs | 40 | 40 | 40 | 40 | 20 | 40 | 50 | `steps_per_epoch` |
| Number of train steps | 50 | 50 | 50 | 1 | 50 | 100 | 100 | `num_train_steps` |
| Replay Buffer size | $1e^6$ | $1e^6$ | $1e^6$ | $1e^6$ | $1e^6$ | $1e^6$ | $1e^6$ | `buffer_size` |
| *Algorithm Settings* | | | | | | | | |
| Policy hidden sizes (MLP) | $(256, 256)$ | $(256, 256)$ | $(256, 256)$ | $(256, 256)$ | $(256, 256)$ | $(256, 256)$ | $(256, 256)$ | `pol_hidden_sizes` |
| Policy noise clip | 0.5 | 0.5 | 0.5 | 0.5 | 0.5 | 0.5 | 0.5 | `pol_noise_clip` |
| Policy noise | 0.2 | 0.2 | 0.2 | 0.2 | 0.2 | 0.2 | 0.2 | `pol_noise` |
| Target update tau | 0.005 | 0.005 | 0.005 | 0.005 | 0.005 | 0.005 | 0.01 | `tau` |
| Buffer batch size | 100 | 100 | 100 | 100 | 100 | 100 | 256 | `batch_size` |
| *Gaussian noise Exploration Settings* | | | | | | | | |
| Max sigma | 0.1 | 0.1 | 0.1 | 0.1 | 0.1 | 0.1 | 0.1 | `max_sigma` |
| Min sigma | 0.1 | 0.1 | 0.1 | 0.1 | 0.1 | 0.1 | 0.1 | `min_sigma` |
| *Optimizer Settings* | | | | | | | | |
| Policy Learning rate | $1e^{-3}$ | $1e^{-3}$ | $1e^{-4}$ | $3e^{-4}$ | $1e^{-3}$ | $3e^{-4}$ | $1e^{-3}$ | `pol_lr` |
| QF Learning rate | $1e^{-3}$ | $1e^{-3}$ | $1e^{-3}$ | $1e^{-3}$ | $1e^{-3}$ | $1e^{-3}$ | $1e^{-3}$ | `qf_lr` |
| *IER Specific Settings* | | | | | | | | |
| Use Hindsight for storing states | − | − | − | ✓ | − | − | − | `use_hindsight` |
| Mixing Factor (p) | 0 | 0 | 0.4 | 0 | 0.3 | 0.8 | 0 | `p` |

Additionally, we use Tabular MDPs for learning a policy in our toy example. Since the environment is fairly simpler, and has very few states, function approximation is unnecessary. For all the agents trained on GridWorld, we use a common setting as described in Table 9.

Table 9: Hyperparameters used for training Tabular MDP on GridWorld-1D environment.

| Description | GridWorld | argument_name |
|---|---|---|
| Discount | 0.99 | `discount` |
| Batch size | 1 | `batch_size` |
| Number of epochs | 100 | `n_epochs` |
| Replay Buffer size | $3e^4$ | `buffer_size` |
| Buffer batch size | 64 | `batch_size` |
| Exploration factor | 0.3 | `max_epsilon` |
| Learning rate | 0.1 | `lr` |

## C   ADDITIONAL RESULTS

### C.1   PERFORMANCE OF IER WITH LOW-DIMENSIONAL STATE SPACE

This section briefly discusses our results on environments with a low-dimensional state space, such as classic control environments (CartPole, Acrobot, and Pendulum) and Box-2D environments (LunarLander). Figure 4 depicts the learning curves of our DQN/DDPG agents in these environments.

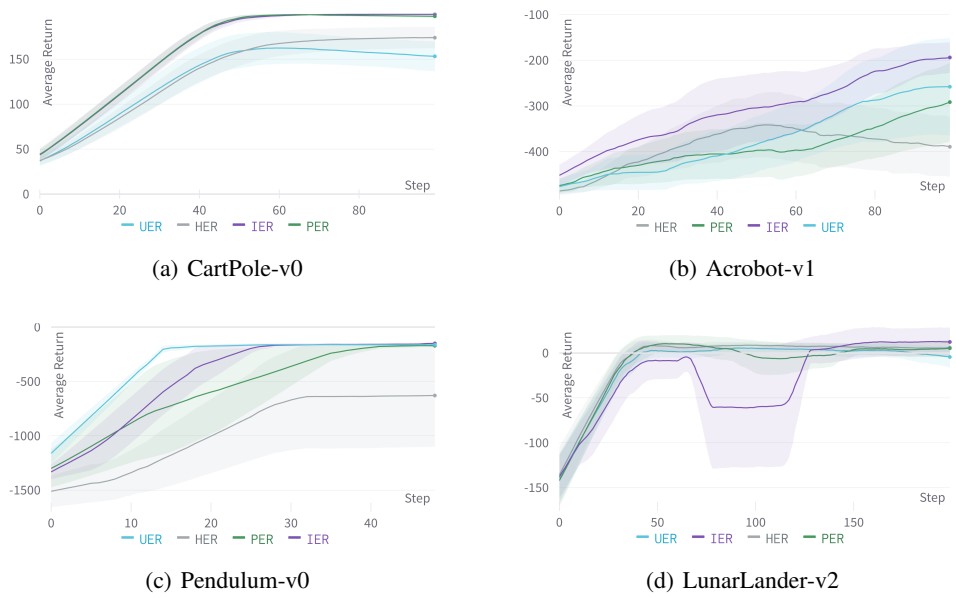

(a) CartPole-v0

(b) Acrobot-v1

(c) Pendulum-v0

(d) LunarLander-v2

Figure 4: Learning curves of DQN/DDPG agents on Classic Control and Box-2D environments.

We note that our proposed methodology can significantly outperform other baselines for the classic control algorithms. Furthermore, Figure 4(a) shows excellent promise as we achieve a near-perfect score across all seeds in one-tenth of the time it took to train PER.

### C.2   PERFORMANCE IN MULTIPLE JOINT DYNAMICS SIMULATION AND ROBOTICS ENVIRONMENTS

Multiple joint dynamic simulation environments (mujoco physics environments) and robotics environments such as HalfCheetah, Ant, Inverted Double-Pendulum, and FetchReach (Todorov et al. (2012); Plappert et al. (2018)) are more complex and enable us to study whether the agent can understand the physical phenomenon of real-world environments. Figure 5 depicts the learning curves of our TD3 agents in these environments. Again, our proposed methodology outperforms all other baselines significantly in most of the environments studied in this section. Additionally, it is essential to point out that our proposed method shows an impressive speedup of convergence in Inverted Double Pendulum and convergence to a much better policy in Ant.

### C.3   PERFORMANCE OF IER IN HUMAN CHALLENGING ENVIRONMENTS

This section briefly discusses our results on human-challenging environments such as Atari environments (Pong and Enduro). These environments are highly complex, and our algorithms take millions of steps to converge to a locally optimal policy. Figure 6 depicts the learning curves of our DQN agents in these environments. We note that our proposed methodology can perform favorably when compared to other baselines for the Atari environments and can reach large reward policies significantly faster than UER.

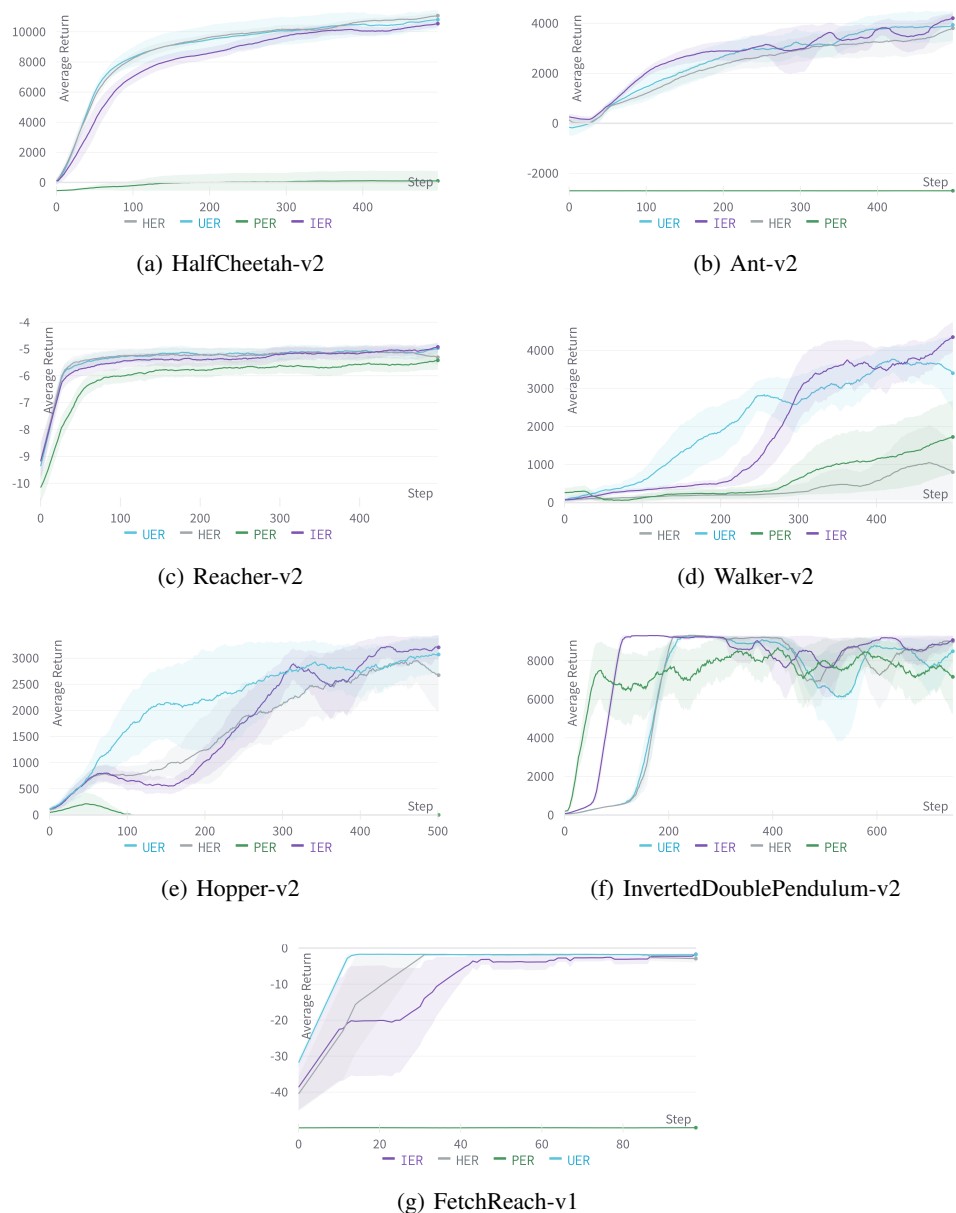

Figure 5: Learning curves of TD3 agents on Mujoco and Robotics environments.

## C.4 WHOLE VS. COMPONENT PARTS

This section briefly presents the learning curves of our models on three different sampling schemes: IER, OER and RER.

## C.5 BUFFER BATCH SIZE SENSITIVITY OF IER

This section briefly presents the sensitivity to the buffer batch size hyperparameter for our proposed approach (IER). To analyze this, we run our experiments on the CartPole environment with varying batch size of the range 2-256. Table 10 and Figure 8 depict the buffer batch size sensitivity results from our proposed sampler.

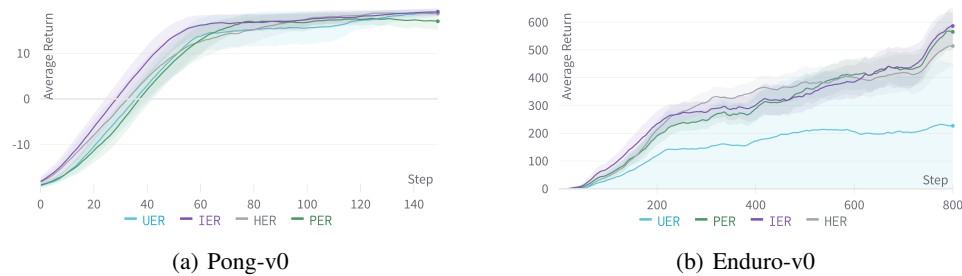

(a) Pong-v0      (b) Enduro-v0

Figure 6: Learning curves of DQN agents on Atari environments.

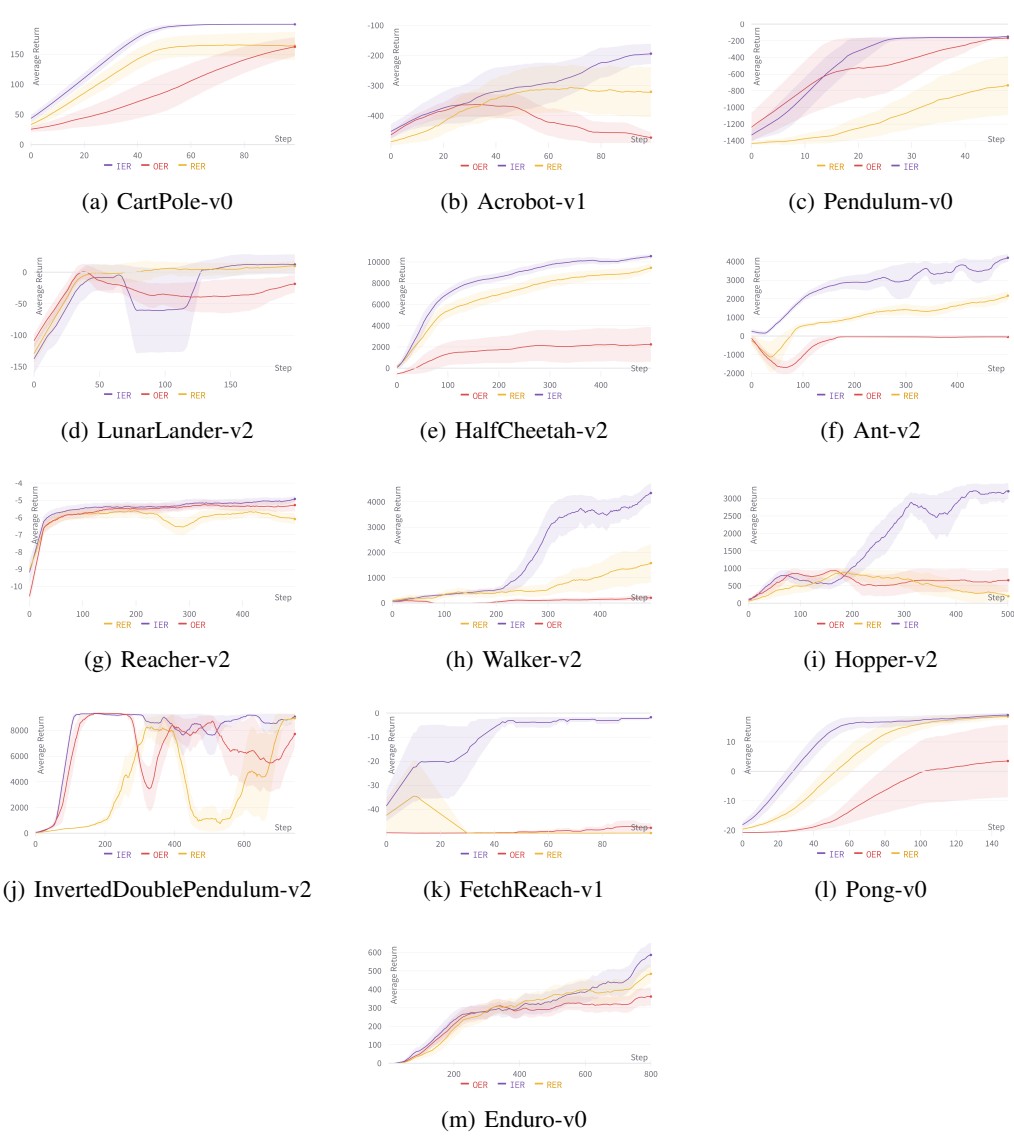

(a) CartPole-v0    (b) Acrobot-v1    (c) Pendulum-v0

(d) LunarLander-v2    (e) HalfCheetah-v2    (f) Ant-v2

(g) Reacher-v2    (h) Walker-v2    (i) Hopper-v2

(j) InvertedDoublePendulum-v2    (k) FetchReach-v1    (l) Pong-v0

(m) Enduro-v0

Figure 7: Ablation study of OER, RER and IER.

Table 10: Buffer Batch size sensitivity of IER on the CartPole environment.

| Buffer Batch Size | Average Reward |
|---|---|
| 2 | $126.69_{\pm 41.42}$ |
| 4 | $192.33_{\pm 13.29}$ |
| 8 | $181.27_{\pm 32.13}$ |
| 16 | $199.24_{\pm 1.32}$ |
| 32 | $\mathbf{199.99}_{\pm 0.001}$ |
| 64 | $199.83_{\pm 0.31}$ |
| 128 | $193.23_{\pm 10.08}$ |
| 256 | $179.95_{\pm 18.94}$ |

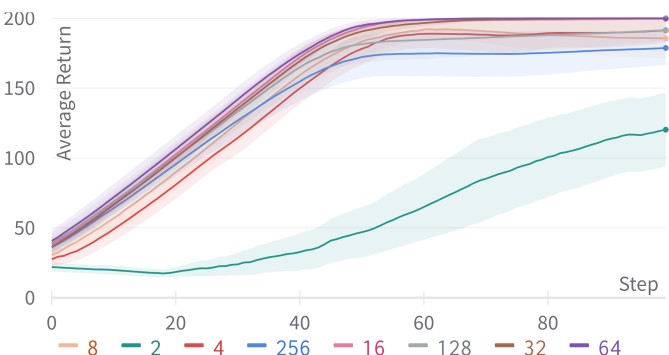

Figure 8: Buffer batch size sensitivity of IER sampler on the CartPole Environment

## C.6 HOW IMPORTANT IS SAMPLING PIVOTS?

This section briefly presents the ablation study to analyze the importance of sampling "surprising" states as pivots. As a baseline, we build a experience replay where these pivots are randomly sampled from the buffer. The "looking back" approach is used to create batches of data. For nomenclature, we refer to our proposed approach (IER) to use the "TD Metric" sampling of pivots, and the baseline that uses "Uniform" sampling of pivots. Table 11 and Figure 9 depict the buffer batch size sensitivity results from our proposed sampler.

Table 11: Importance of sampling pivots in our proposed approach (IER) on the CartPole environment.

| Sampling Scheme | Average Reward |
|---|---|
| TD Metric (IER) | $\mathbf{199.83}_{\pm 0.31}$ |
| Uniform (IER) | $136.71_{\pm 19.59}$ |

## C.7 HOW IMPORTANT IS "LOOKING BACK"?

This section briefly presents the ablation study to analyze the importance of "looking back" after sampling pivots. As a baseline, we build a experience replay where we sample uniformly instead of looking back. For nomenclature, we refer to our proposed approach (IER) to use the "Looking Back" approach (similar to IER), and the baseline that uses "Uniform" approach. We refer to these two approaches as possible filling schemes, i.e. fill the buffer with states once the pivot state is sampled.

Table 12 and Figure 10 depict the buffer batch size sensitivity results from our proposed sampler.

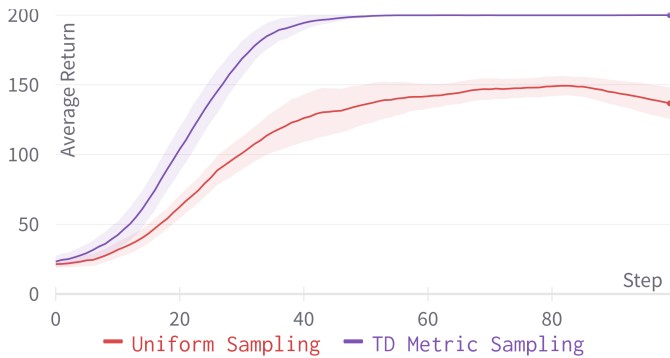

Figure 9: Ablation study of Importance Sampling of IER sampler on the CartPole Environment. Here "Uniform Sampling" denotes the unbiased and random sampling of pivots, and "TD Metric Sampling" denotes our proposed approach (IER).

Table 12: Importance of looking back in our proposed approach (IER) on the CartPole environment.

| Filling Scheme | Average Reward |
|---|---|
| Looking Back (IER) | **199.83** $_{\pm 0.31}$ |
| Uniform (IER) | 182.5 $_{\pm 23.49}$ |

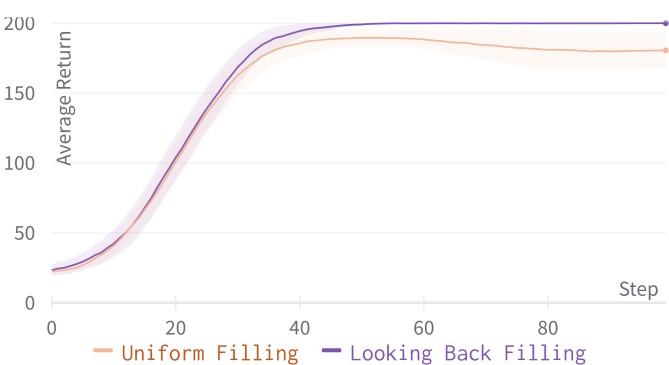

Figure 10: Ablation study of Filling Scheme of IER sampler on the CartPole Environment. Here "Uniform Filling" denotes the unbiased and random sampling of states to fill after sampling the pivot state, and "Looking Back Filling" denotes our proposed approach (IER).

## D  ABLATION STUDY OF TEMPORAL EFFECTS

This section studies the ablation effects of going temporally forward and backward once we choose a pivot/surprise point. Furthermore, Figure 11 depicts the learning curves of the two proposed methodologies. We notice that against theoretical intuition, the unbiased forward sampling scheme is worse in most environments compared to the reverse sampling scheme.

## E  SPARSITY AND REWARDS OF SURPRISING STATES

### E.1  SURPRISING STATES HAVE LARGE REWARDS

In this section, we study the "learning from sparse reward" intuition provided in Section 4 – i.e., we want to check if the states corresponding to large TD error correspond to states with large (positive or

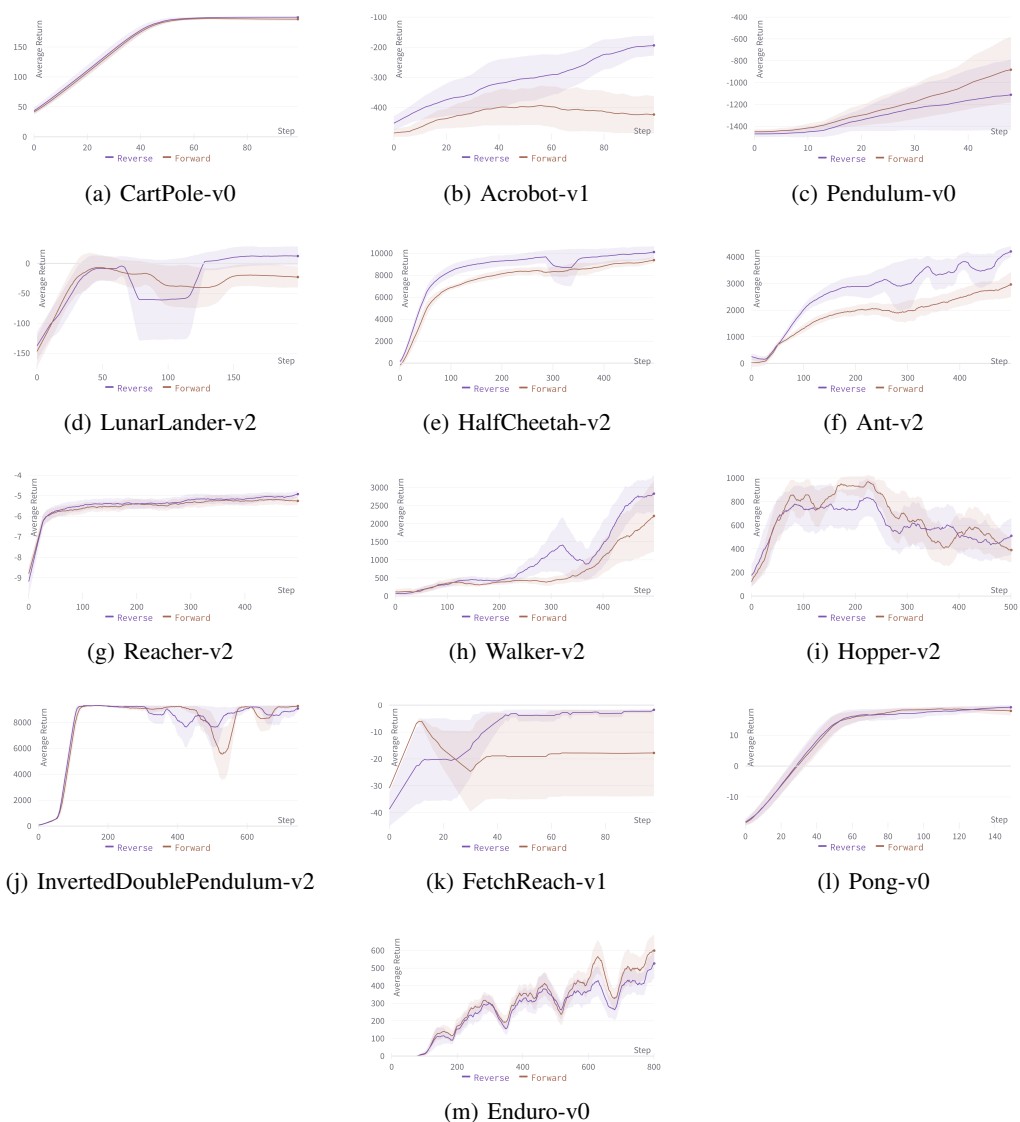

Figure 11: Ablation study of the effects of the temporal structure on the performance of the agent.

negative) rewards. To test the hypothesis, we consider a sampled buffer and plot the TD error of these points in the buffer against the respective reward. Figure 3 shows the distribution of TD error against reward for the sampled buffers in the Ant environment. We see that high reward states (positive or negative) also have higher TD errors. Therefore, our algorithm picks large reward states as endpoints to learn in such environments.

### E.2 SURPRISING STATES ARE SPARSE AND ISOLATED

Figure 12 and Figure 13 depict the distribution of "surprise"/TD error in a sampled batch for CartPole and Ant environments respectively. These two figures help show that the states with a large "surprise" factor are few and that even though the pivot of a buffer has a large TD error, the rest of the buffer typically does not.

Figure 12(d) and Figure 13(d) show a magnified view of Figure 12(c) and Figure 13(d) where the pivot point selected is dropped. This helps with a uniform comparison with the remaining timesteps within

the sampled buffer. Again, we notice little correlation between the timesteps within the sampled buffer.

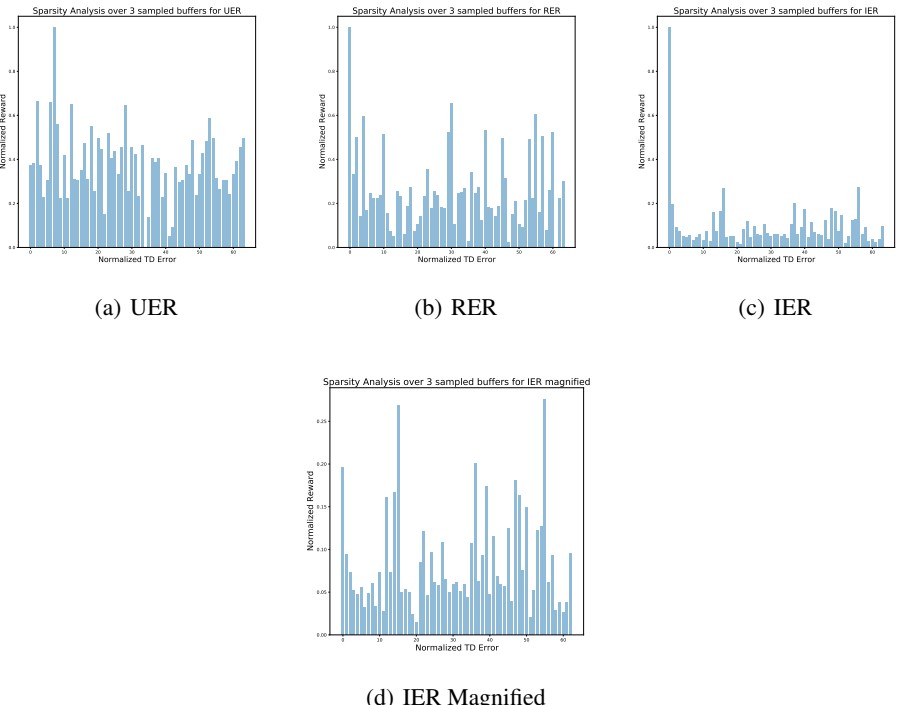

(a) UER        (b) RER        (c) IER

(d) IER Magnified

Figure 12: Normalized TD Error ("Surprise factor") of each timestep over three different sampled buffers on the CartPole environment. Best viewed when zoomed.

## F  REVERSE EXPERIENCE REPLAY (RER)

This section discusses our implementation of Reverse Experience Replay (RER), which served as a motivation for our proposed approach. The summary of the RER approach is shown in Figure 14. Furthermore, an overview of our implemented approach to RER is described briefly in Algorithm 2.

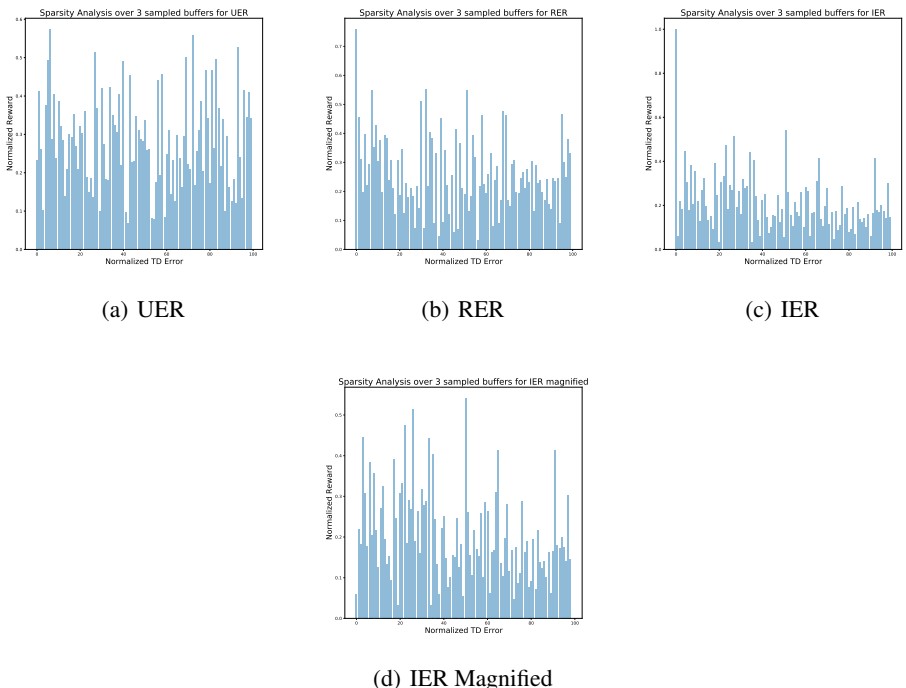

(a) UER  (b) RER  (c) IER

(d) IER Magnified

Figure 13: Normalized TD Error ("Surprise factor") of each timestep over three different sampled buffers on the Ant environment. Best viewed when zoomed.

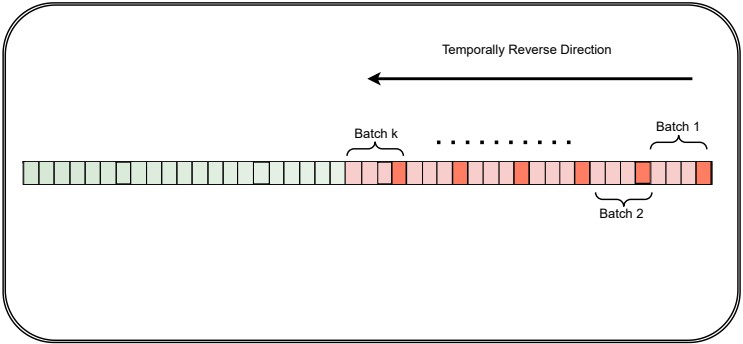

Figure 14: An illustration of Reverse Experience Replay (RER) when selecting $k$ batches from the Replay Buffer.

**Algorithm 2:** Reverse Experience Replay

**Input:** Data collection mechanism $\mathbb{T}$, Data buffer $\mathcal{H}$, Batch size $B$, grad steps per Epoch $G$,
number of episodes $N$, learning procedure $\mathbb{A}$

$n \leftarrow N$;
$P \leftarrow \mathsf{len}(\mathcal{H})$ ;                    // Set index to last element of Buffer $\mathcal{H}$
**while** $n < N$ **do**
  $n \leftarrow n + 1$;
  $\mathcal{H} \leftarrow \mathbb{T}(\mathcal{H})$ ;                    // Add a new episode to the buffer
  $g \leftarrow 0$;
  **while** $g < G$ **do**
    **if** $P - B < 0$ **then**
      $P \leftarrow \mathsf{len}(\mathcal{H})$ ;        // Set index to last element of Buffer $\mathcal{H}$
    **else**
      $P \leftarrow P - B$;
    **end**
    $D \leftarrow \mathcal{H}[P - B, P]$ ;      // Load batch of previous $B$ samples from
    index $P$
    $g \leftarrow g + 1$;
    $\mathbb{A}(D)$;        // Run the learning algorithm with batch data $D$
  **end**
**end**

