# OpenReview forum: "Look Back When Surprised: Stabilizing Reverse Experience Replay for Neural Approximation"
_ICLR.cc/2023/Conference — Submitted to ICLR 2023_

### Official Review · Reviewer_HzYs · 2022-10-24

**Confidence:** 3
**Clarity, Quality, Novelty And Reproducibility:** I comment on each of these in the abo…
**Correctness:** 3
**Technical Novelty And Significance:** 2
**Empirical Novelty And Significance:** 2
**Recommendation:** 6

**Strength And Weaknesses:**

Strength:

1. The topic of ER mechanism is interesting and important;
2. The paper presents its mean idea very clear;
3. The proposed sampling method seems to be novel;
4. The paper conduct extensive experiments on a variety of domains.

Weaknesses:

---------------------------
Algorithm design. The presented algorithm 1 has to collect data episode by episode. This turns the algorithm into an offline algorithm, restricting its utility. But it seems the sampling method can be done in an online manner, why not propose that? This is important because 1) an online method is a closer competitor to PER which runs online (update parameters at each environment time step) and 2) it is clearer how the two algorithms are compared.

Figure 3 is not persuasive. The TD errors can change as the parameters get updated. I do not see a clear correlation between reward magnitude and TD error magnitude. Also, if this correlation is true and is beneficial, shouldn't PER perform very well in the sparse reward setting?

---------------------------
Concerns about experiments. The empirical results are extensive but not persuasive.

1. Many figures (Fig 4-6) in the experiments section include learning curves with very large variances/standard errors, where one cannot really identify the proposed method to be better than others. Furthermore, it is better to study the hyper-parameter sensitivity of the proposed algorithm. The algorithm seems to have a large reliance on the size of the ER buffer.

2. Missing at least two intuitive baselines to make the proposed method more persuasive:
1). Uniformly sample the pivot points, and the rest is the same as the proposed method: this can verify the usefulness of the claimed "reverse replay."
2). Prioritized sampling of the pivot points and then uniform sampling of the rest of datapoints in each batch: this can further validate the reverse sampling is important

3. One critical question about the experiments. The PER/UER can update parameters at each time step, while RER++ needs to wait until the end of an episode. How do you conduct the comparison? Do you use the same amount of real environment data or the same computation power for all algorithms?

4. Missing details of the PER. PER has a mechanism to anneal the sampling bias. Since I see the proposed method used a "mixed replay" method to mitigate bias, it is important to report if there any effort (tuning the hyper-parameter) of the PER baseline has been made to anneal the bias.


---------------------------
Missing related work. The paper belongs to the broad subarea of the sampling distribution of experiences, and there are many more papers in this category that should be discussed. I name a few highly relevant works here:
[1] An equivalence between loss functions and non-uniform sampling in experience replay by Scott Fujimoto et al.
[2] Remember and forget for experience replay by Guido Novati et al.
[3] Understanding and mitigating the limitations of PER by Yangchen Pan et al.
[4] regret minimization ER in off-policy RL et al.

All these papers discuss the pros and cons of PER/ER methods, and some of them shed light on the theoretical mechanisms behind why a sampling method should be beneficial or what a good sampling distribution should be.

----------------------
Presentation issue (I consider this not critical, but it can be significantly improved). The proposed sampling approach is not well-motivated. In the abstract, it says PER and UER may suffer from large bias and sub-optimal convergence, respectively. However, there is no evidence in the paper showing the proposed IER method is optimal or has a small bias. In fact, PER does address the biased sampling issue by using an important ratio, as introduced in the original paper (section 3.4). In contrast, the proposed method does not even have a sound method to anneal the sampling bias.
Furthermore, the paper attempts to use RER to motivate their approach too. However, the RER theory (from a system identification setting or linear MDP setting) cited by the authors does not really apply to general RL settings. I do not mean the authors need to provide a new/strong theory to motivate their method. Still, it would be a plus if the authors specify which theorems from existing work motivate their algorithmic design.


**Summary Of The Paper:**

The paper studies experience replay mechanisms in a deep reinforcement learning (DRL) setting. Notably, it proposes a modification to the existing reverse experience replay method. The paper claims that purely prioritizing experiences according to TD errors and a naive (uniform) ER method may suffer from sub-optimal convergence and potentially large bias, respectively. Then the paper proposes the following method. First, the algorithm picks up k “pivot points” from a large buffer according to the TD error-based sampling distribution. Second, get the corresponding k batches of data where each batch ends with a pivot point sampled in the previous step. Then those mini-batches are used to update the training parameters. Extensive experiments on simple discrete domains, mujoco domains, and Atari games are conducted to show the effectiveness of the algorithms.


**Summary Of The Review:**

The paper studies an important topic and presents a new ER method with extensive empirical results. However, I think the drawbacks of the algorithmic and experimental design currently outweigh the advantages. I will adjust my score based on reading the author's response and other reviewers' opinions.

---

> ### Author Response · Authors · 2022-11-08
> **A Response to the Reviewer**
>
> We thank the reviewer for all the insightful comments. These have helped us make our work better in various ways. Please refer to the common comments where we have introduced more experiments based on reviewer feedback, taking the total to 13. We are working to improve the presentation issues but have addressed all the major technical comments already in the revisions. We have responded to the reviewer's comments below.
>
> ---
>
> - *“Algorithm design. The presented algorithm 1 has to collect data episode by episode. This turns the algorithm into an offline algorithm, restricting its utility. But it seems the sampling method can be done in an online manner, why not propose that? This is important because 1) an online method is a closer competitor to PER which runs online (update parameters at each environment time step) and 2) it is clearer how the two algorithms are compared."*
>
> **AND**
>
> - *"One critical question about the experiments. The PER/UER can update parameters at each time step, while RER++ needs to wait until the end of an episode. How do you conduct the comparison? ”*
>
> There is some confusion with the terminology here, which we will first clarify. In offline RL (as defined in the literature [1]), the agent only has access to offline data, usually from many episodes. This setting does not allow one to deploy and query results for new policies. Therefore, our algorithm is online since we actively interact with the environment and deploy new policies each episode based on the data collected. **Therefore, we disagree that our method is offline**.
> As we understand it, the reviewer's question is regarding the update/training taking place after a fixed number of steps or after each episode - which is an aspect of the DQN/ TD3 type RL algorithms rather than the ER method itself. We note that even in Deepmind’s original DQN code, the update happens after a fixed number of steps (a specified hyperparameter) and not after each step ([code here](https://github.com/deepmind/dqn)).
> The reviewer is correct that we can easily modify the IER method to collect data step-by-step and for the RL algorithm like DQN/ TD3 to update step-by-step. Many standard implementations of RL algorithms ([2], [3], [4], [5],[6]) and many others that use [3] as a backbone (similar to us) add entire episodes into the replay buffer before updating the Q network/ policy network. Furthermore, adding episode by episode ([link to specific code file](https://github.com/rlworkgroup/garage/blob/3492f446633a7e748f2f79077f6301c5b3ec9281/src/garage/torch/algos/dqn.py#L192)) does not affect the agents' performance since these implementations can reproduce the results of the original papers. Indeed, these codebases are widely used by RL researchers. This is the reason we compared the episode-by-episode versions for all replay methods. Note that the authors of the PER paper have not published the original code.
>
> [1] Offline Reinforcement Learning: Tutorial, Review, and Perspectives on Open Problems. Levine et al.
>
> [2] “GitHub - Rlcode/per: Prioritized Experience Replay (PER) Implementation in PyTorch.” GitHub, github.com/rlcode/per
>
> [3] rlworkgroup. “GitHub - Rlworkgroup/Garage at per_Dqn.” GitHub, github.com/rlworkgroup/garage/tree/per_dqn
>
> [4] gxywy. “GitHub - Gxywy/Pytorch-dqn: A Simple Implementation of Dqn Algorithm Using Pytorch.” GitHub, github.com/gxywy/pytorch-dqn.
>
> [5] transedward. “Pytorch-dqn/dqn_learn.py at Master · Transedward/Pytorch-dqn.” GitHub, github.com/transedward/pytorch-dqn.
>
> [6] AndersonJo. “GitHub - AndersonJo/Dqn-pytorch: Deep Q Learning via Pytorch.” GitHub, github.com/AndersonJo/dqn-pytorch.

---

> > ### Author Response · Authors · 2022-11-08
> > **Reponse to the Review (Continued)**
> >
> > - *"Figure 3 is not persuasive. The TD errors can change as the parameters get updated. I do not see a clear correlation between reward magnitude and TD error magnitude. Also, if this correlation is true and is beneficial, shouldn't PER perform very well in the sparse reward setting?"*
> >
> > This is indeed possible. However, our intuition does hold in the initial part of the training when the initial, naive policies find it hard to reach high-reward states often. Even when the policy is good but far from optimal, it might rarely reach the highest-reward states. Our explanation is that this gives a significant advantage to IER in learning from these few high-reward states compared to UER. In the paper's revised version, we plotted normalized |reward| vs. |TD error|. We also fit a straight line to this plot to make the correlation visible. We hope the reviewer is satisfied with this plot.
> > PER is also expected to perform well, but can suffer from a considerable bias error (without careful hyperparameter tuning) by sampling *mostly* high TD error samples. Our method considers the immediate predecessor states of the high TD error states, which do not seem to have high TD error (See figures 12 and 13 in the Appendix). Based on our discussion (Section 3), IER learns the trajectory which reached the high reward states. PER does not have this.
> >
> > ---
> >
> > - *"Many figures (Fig 4-6) in the experiments section include learning curves with very large variances/standard errors, where one cannot really identify the proposed method to be better than others."*
> >
> > You are correct that a few learning curves in the experiments section have a large variance. However, this phenomenon is common in RL, as shown by well-known works such as [7], [8], [9]. This is not a phenomenon observed for IER alone. To accurately identify the better performance of IER over other baselines, please refer to Table-1. Our IER algorithm outperforms all other baselines in 11/13 environments (from a wide class of environments such as classic control, box3d, mujoco, robotics, as well as atari). It is in Top-2 in our remaining experiments.
> >
> > [7] Henderson, Peter, et al. "Deep reinforcement learning that matters." Proceedings of the AAAI conference on artificial intelligence. Vol. 32. No. 1. 2018.
> >
> > [8]Rajeswaran, Aravind, Igor Mordatch, and Vikash Kumar. "A game theoretic framework for model based reinforcement learning." International conference on machine learning. PMLR, 2020.
> >
> > [9] Schulman, John, et al. "Proximal policy optimization algorithms." arXiv preprint arXiv:1707.06347 (2017).
> >
> > ---
> >
> > - *"Furthermore, it is better to study the hyper-parameter sensitivity of the proposed algorithm. The algorithm seems to have a large reliance on the size of the ER buffer."*
> >
> >  It is unclear to us which experiments the reviewer used to arrive at this conclusion. We have run experiments on the CartPole environment using IER sampler, and confirm that the algorithm does not seem to have a “large” reliance on the size of the ER buffer, unless we experiment with extreme batch sizes such as 2,4, etc which are seldom used in practice. As suggested, we have added this study to the Appendix of our revised draft. **We reiterate that we have used the same buffer size for all ER algorithms**.
> >
> >
> > > | Sampler Buffer Size | Average Reward |
> > |---|---|
> > | 2 | 126.688 ± 41.417 |
> > | 4 | 192.328 ± 13.288 |
> > | 8 | 181.27 ± 32.131 |
> > | 16 | 199.238 ± 1.319 |
> > | 32 | 199.99 ± 0.001 |
> > | 64 | 199.829 ± 0.3057 |
> > | 128 | 193.228 ± 10.075 |
> > | 256 | 179.948 ± 18.936 |
> >
> > ---
> >
> > - *"Missing at least two intuitive baselines to make the proposed method more persuasive: 1). Uniformly sample the pivot points, and the rest is the same as the proposed method: this can verify the usefulness of the claimed "reverse replay." 2). Prioritized sampling of the pivot points and then uniform sampling of the rest of datapoints in each batch: this can further validate the reverse sampling is important"*
> >
> > Thank you for pointing these out. Sampling uniformly and looking back was tried by us during the first phase of this project. It turned out to be a poor baseline; therefore, we excluded it from our current draft for clarity. Below, we showcase the baseline on the simple Cartpole environment. This shows that the TD metric being used is essential.
> >
> > > | Experience Replay | Average Reward |
> > |:---:|:---:|
> > | Pivot + “Looking Back” (IER) | 199.83 ± 0.31 |
> > | Uniform + “Looking Back” | 136.708 ± 19.59 |

---

> > > ### Author Response · Authors · 2022-11-08
> > > **Response to the Reviewer (Continued)**
> > >
> > > The second baseline would intuitively be analogous to the uniform experience replay, which has been studied extensively in this section. Furthermore, we confirm this and report below the comparison against the suggested experience replay on the CartPole environment. Again, as suggested, we add this study to the Appendix of our revised draft.
> > >
> > > > | Experience Replay | Average Reward |
> > > |:---:|:---:|
> > > | “Looking Back” + Pivot (IER) | 199.83 ± 0.31 |
> > > | Uniform + Pivot | 182.5 ± 23.49 |
> > >
> > > We detail the below results along with their learning curves in the updated draft (See Figures 10 and 11 in the Appendix).
> > >
> > > ---
> > >
> > > - *"Do you use the same amount of real environment data or the same computation power for all algorithms?"*
> > >
> > > Yes, all our experiments use the same amount of data, and computation power (gpus) as described in the appendix.
> > >
> > > ---
> > >
> > > - *"Missing details of the PER. PER has a mechanism to anneal the sampling bias. Since I see the proposed method used a "mixed replay" method to mitigate bias, it is important to report if there any effort (tuning the hyper-parameter) of the PER baseline has been made to anneal the bias."*
> > >
> > >  In [Schaul et al., 2015], the authors confirmed using a common hyperparameter across all the diverse 50 environments of atari. These are the commonly used hyper-parameter settings for PER in the codebases referred to above. Since these hyper-parameters were found after an extensive search, we use the prescribed hyperparameters.
> > >
> > > ---
> > >
> > > - *"All these papers discuss the pros and cons of PER/ER methods, and some of them shed light on the theoretical mechanisms behind why a sampling method should be beneficial or what a good sampling distribution should be."*
> > >
> > > Thank you for your input, we have currently added these relevant works mentioned in our revised version.
> > >
> > > ---
> > >
> > > - *"The proposed sampling approach is not well-motivated."*
> > >
> > > We respectfully disagree with this comment. We have made a lot of effort to motivate our approach.
> > >
> > > 1. The method is based on the intuition that an RL method needs to learn the cause for surprising actions by looking back.
> > >
> > > 2. We have described several theoretical works considering reverse replay methods and empirical works considering reverse sweep techniques.
> > >
> > > 3. We have also demonstrated a didactic toy example where we explain why IER can outperform other methods by looking back from rare-surprising states more often.
> > >
> > > 4. Our empirical results demonstrate the superiority of our technique.
> > >
> > > ---
> > >
> > > - *"In the abstract, it says PER and UER may suffer from large bias and sub-optimal convergence, respectively. However, there is no evidence in the paper showing the proposed IER method is optimal or has a small bias."*
> > >
> > > We respectfully disagree with this comment. We claim that IER is the optimal method and has a small bias among the ER techniques based on the results of 13 experiments, of which IER is the best in 11. Similarly, if IER had a large bias, it would not have performed so well on most tasks. Moreover, IER method in the didactic toy example has better and more uniform coverage over the entire state space than the other methods.
> > >
> > > ---
> > >
> > >  - *"In fact, PER does address the biased sampling issue by using an important ratio, as introduced in the original paper (section 3.4). In contrast, the proposed method does not even have a sound method to anneal the sampling bias. "*
> > >
> > > We respectfully disagree with this comment. The comment “proposed method does not even have a sound method to anneal the sampling bias”  seems unjustified given that we have a dedicated parameter (p) that can reduce bias by mixing it with UER. However, we set p = 0 most of the time, which shows that our method inherently has low bias. We refer to the discussion in Section 6 for further details.
> > > The fact that IER has inherently low bias is demonstrated by the fact that in the didactic toy example, where IER removes the bias present in OER. Also, given that we have demonstrated the efficacy of our techniques in Table-2, the comment made by the reviewer is confusing.

---

> > > > ### Author Response · Authors · 2022-11-08
> > > > **Response to Reviewer (Continued)**
> > > >
> > > > - *"Furthermore, the paper attempts to use RER to motivate their approach too. However, the RER theory (from a system identification setting or linear MDP setting) cited by the authors does not really apply to general RL settings. I do not mean the authors need to provide a new/strong theory to motivate their method. Still, it would be a plus if the authors specify which theorems from existing work motivate their algorithmic design."*
> > > >
> > > > We claim that RER is theoretically rigorous in the linear function approximation setting (including the important tabular MDPs). We refer to Theorems 1 and 2 in (Agarwal et al., 2021) for the exact results for RER - here, the authors obtain the first optimal convergence guarantees for tabular MDPs which was not possible with forward pass Q learning before. We note that, to the best of our knowledge, other forms of experience replay have not even been rigorously analyzed in this simple setting with any convergence guarantees. Infact, prior theoretical works make the unsubstantiated assumption that using UER means obtaining i.i.d. samples from some stationary distribution (see [10],[11] below)
> > > >
> > > > [10] A new convergent variant of Q-learning with linear function approximation, Carvalho et al 2020
> > > > [11] A Theoretical Analysis of Deep Q-Learning, Fan et al, 2020
> > > >
> > > > RL algorithms (like Q learning, policy gradients, policy iteration etc.) have been traditionally designed to work in the tabular/linear setting (rigorously) and then extended to more complex scenarios with non-linear function approximation, without the same theoretical guarantees. To the best of our knowledge, no satisfactory theoretical analysis describes any RL algorithm’s workings in the neural function approximation setting. Therefore, such an analysis is beyond the scope of this work.
> > > >
> > > > ---
> > > >
> > > > We appreciate your suggestions and are pleased to know that you find the problem not only interesting and important, but also appreciate the fact that our proposed methodology is novel, clear and outperforms other baselines. Given the fact that we have answered all of the concerns raised by the reviewer, we hope you are willing to bump our score to acceptance.
> > > > Please do get back to us if there are any further corrections that could help polish our paper. Again, thank you for your time and effort in reviewing our work!

---

> > > > ### Comment · Reviewer_HzYs · 2022-11-23
> > > > **thanks for the response**
> > > >
> > > > Your detailed response does address some of my concerns. In general, I reconsidered some of my concerns and believe that they are not critical. I will raise my score later.
> > > >
> > > > 1. Regarding the biased sampling issue.
> > > >
> > > > By saying "the method does not have a solid way to remove bias," what I mean is you did not show E[proposed estimator] = E[unbiased estimator].
> > > >
> > > > The PER paper does use an important ratio to establish equality, although their actual implementation provides an additional mechanism. Of course, whether this equality holds also depends on how you define the unbiased sample. But I do not see a clear description in this paper.
> > > >
> > > > 2. "In the abstract, it says PER and UER may suffer from large bias and sub-optimal convergence, respectively. However, there is no evidence in the paper showing the proposed IER method is optimal or has a small bias."
> > > >
> > > > Simply saying that your algorithm does better in many experiments only provides suggestive, but not conclusive evidence regarding the "optimality" of your algorithm.
> > > >
> > > > By claiming an algorithm converges to optimal, the first thing is to mathematically describe/define what the optimum is and then show the convergence proof. I would not use such a strong claim to assert the benefit of an algorithm without provable evidence.
> > > >
> > > > By claiming a sample is unbiased, as I mentioned above, there is a clear statistical definition of biasness given what the unbiased sample/population parameter is. Showing unbiasedness should be done by showing that equality holds.
> > > >
> > > > Regarding the empirical evidence. As the authors agree, some learning curves have large STE. Furthermore, I personally have lots of experience running PER. The algorithm can be actually quite sensitive to the bias annealing parameter. Even doing a slight tuning can provide nontrivial improvement in some domains, especially those large domains.
> > > >
> > > > I want to emphasize that the motivation part is more like a presentation issue, and is not critical in deciding the score. ER is a popular research topic and there are many existing variants, I do think carefully positioning the proposed method among so many existing ER-related methods could significantly increase the chance of acceptance.

---

> > > > > ### Author Response · Authors · 2022-11-25
> > > > > **Response to Reviewer (Part 2)**
> > > > >
> > > > > Thank you for your response. We highly appreciate the additional questions you have raised. Please find below our response to your following queries.
> > > > >
> > > > > ---
> > > > >
> > > > > - *"By saying "the method does not have a solid way to remove bias," what I mean is you did not show E[proposed estimator] = E[unbiased estimator]. The PER paper does use an important ratio to establish equality, although their actual implementation provides an additional mechanism. Of course, whether this equality holds also depends on how you define the unbiased sample. But I do not see a clear description in this paper."*
> > > > >
> > > > > **AND**
> > > > >
> > > > > - *"By claiming a sample is unbiased, as I mentioned above, there is a clear statistical definition of biasness given what the unbiased sample/population parameter is. Showing unbiasedness should be done by showing that equality holds."*
> > > > >
> > > > > Thank you for observing the confusing manner in which the word "bias" has been used in the paper. We clarify this below. We will make changes to all these claims, including rewording the term `optimal’ (by which we meant that it is optimal among the choices of ER buffers considered).
> > > > >
> > > > > Concerning the word "bias" - a part of the ambiguity is because we are describing biases at various stages of the algorithm. We did not include explanations due to space constraints but will include them in the next version. Let us explain the multiple usages of this word in this work. The word bias has been used similarly in the PER paper too, as we show below.
> > > > >
> > > > >
> > > > > **The first usage** of the word bias is in the context of the toy model where we mean that `some states are sampled while others are ignored’ by the ER technique. The Bellman operator is an $\ell^{\infty}$ contraction and not an $\ell^2$ contraction, we need to have accurate estimates of the Bellman operator of **all states** to apply the Bellman operator effectively recursively. If our sampling procedure returns some states very rarely (as with uniform sampling from the buffer in the toy example), the convergence of the Q learning procedure can become impeded. This also suggests sampling from a uniform distribution over the state space (**not the same as the uniform distribution over the buffer**). In this case, by bias, we mean the bias with respect to the uniform distribution over the state-space, which ensures $\ell^{\infty}$ contraction. In this sense, IER is 'less biased’ than the uniform distribution over the state space, as demonstrated.
> > > > >
> > > > > “Bias” in this sense has also been used in the PER paper, where they claim that “uniform sampling is implicitly biased toward out-of-date transitions that were generated by a policy that has typically seen hundreds of thousands of updates since” - meaning it does not give samples which are useful to learn from.
> > > > >
> > > > > **The second usage** concerns the bias in the learning algorithm, deployed to approximate the Bellman iteration. We will first distinguish the two kinds of ‘unbiased’ estimators, which can be defined here:
> > > > >
> > > > > a) Unbiased with respect to the sample distribution in the buffer (i.e., (s_t,a_t) is such that t is drawn uniformly from [T], where T is the buffer size).
> > > > > b) Unbiased with respect to the population gradient for given (s, a). The bias here refers to the selection bias of outcomes (which we explain below. That is, sample state-action $(s_t,a_t,r_t,s_t^{\prime})$ at time $t$ in some manner and consider the Q-value updates ($\mathcal{T}$ is the Bellman operator):
> > > > > $$Q_{t+1}(s_t,a_t) = (1-\eta)Q_t(s_t,a_t) + \eta \mathcal{T}(Q^{target}) (s_t,a_t) + \eta \epsilon_t(s_t,a_t,s_t^{\prime},r_t)$$ Here $\epsilon_t(s_t,a_t,s^{\prime}) = r_t +\max_{a} Q^{target}(s^{prime}_t,a) - \mathcal{T}(Q^{target}) (s_t,a_t)$.
> > > > >  Here, the algorithm is unbiased if for any $(s_t,a_t, r_t,s^{prime}_t)$, we have $$E(\epsilon_t(s_t,a_t,s_t^{\prime},r_t)|s_t,a_t] = 0$$
> > > > >  (i.e., our estimate for the bellman operator is unbiased at $(s_t,a_t)$). This implies that the Q-learning procedure converges.
> > > > >
> > > > > Bias in the sense of b) has also been used in the PER paper, where they also admit to this kind of (small) bias: *“In typical reinforcement learning scenarios, the unbiased nature of the updates is most important near convergence at the end of training, as the process is highly non-stationary anyway, due to changing policies, state distributions and bootstrap targets; we hypothesize that a small bias can be ignored in this context (see also Figure 12 in the appendix for a case study of full IS correction on Atari).”*
> > > > >
> > > > > We disagree with the claim *“The PER paper does use an important ratio to establish equality, although their actual implementation provides an additional mechanism”*  - We did not find the proof of equality in the PER paper. It cannot be unbiased in the sense a) or b) as described above. Being unbiased in the sample sense (a)) defeats the purpose of prioritization, and being unbiased in the sense b) cannot happen since the algorithm picks importance by observing the next state and rewards $(s^{\prime}_t,r_t)$.

---

> > > > > > ### Author Response · Authors · 2022-11-25
> > > > > > **Response to Reviewer (Part 2; continued)**
> > > > > >
> > > > > > Showing a) does not help show faster convergence to the global optimum. As the toy model demonstrates, the policy used to collect the buffer in the first place may contain very few samples from important states, which hinders the learning process, as explained above. Note that neither PER nor IER are unbiased in this sense. This is the main reason we choose to deploy techniques like PER and IER.
> > > > > >
> > > > > >  Therefore, our work considers unbiasedness in the sense of Property b). Property b) is shown for RER in (Agarwal et al. 2021). We note that an algorithm can be biased in the sense of a) but still be unbiased in the sense of b). As long as Property b) holds, our algorithm converges due to guarantees for stochastic approximation. In case property b) does not hold, then the algorithm cannot perform well. Therefore, the performance of our algorithm itself is evidence that the algorithmic bias is low.
> > > > > >
> > > > > >
> > > > > > Let us also elaborate on this point further. Given a large buffer of samples, we can pick $(s_t,a_t, r_t, s_t^{\prime})$ using any importance sampling metric which depends on (s_t,a_t) alone, then observe reward and next state $(r_t,s_t^{\prime})$ and achieve the unbiasedness in condition b).
> > > > > > Whenever the importance depends on $s_t^{\prime}$ and/or $r_t$ also, we cannot ensure that $epsilon_t$ has zero conditional mean (since we are selecting outcomes of the current state and action, causing selection bias). To illustrate this, consider the scenario where there is a rare high-reward event (which occurs with a 1% probability). This will be present many times within a large buffer. Sampling such points preferentially (i.e., by looking at the reward along with the state and action) gives a biased estimator for the bellman operator in the Q learning update, which hinders the Q learning updates.
> > > > > >
> > > > > > This problem does not arise in deterministic MDPs (which are common in modern RL, and most of the experiments considered are of deterministic MDPs) where $s_t^{\prime}, r_t$ are deterministic functions of $(s_t,a_t)$. In this case, our algorithm is still unbiased. Referring to the response to reviewer (j7Bo), even in stochastic environments, we can average out the importance metric with respect to $s_t^{\prime},r_t$. That is, given an importance metric $$I(s,a,s^{\prime},r)$$, we construct the new metric:
> > > > > >
> > > > > > $$I_1(s,a) = \mathbb{E}\left[I(s,a,s^{\prime},r)|s,a\right]$$
> > > > > > The RHS can be estimated by an empirical average over the buffer. In the case of continuous state spaces, we can cluster the states based on certain features and then compute the importance of each of the clusters. This is more computationally intense but is expected to work even in stochastic environments. We intend to tackle these questions in future work.
> > > > > >
> > > > > > ---
> > > > > >
> > > > > > - *"Regarding the empirical evidence. As the authors agree, some learning curves have large STE. Furthermore, I personally have lots of experience running PER. The algorithm can be actually quite sensitive to the bias annealing parameter. Even doing a slight tuning can provide nontrivial improvement in some domains, especially those large domains."*
> > > > > >
> > > > > > Thank you for your insights. We have applied a grid-search to select the best parameters of not only the bias annealing parameter, but also the prioritization exponent $\alpha$ in the cartpole and double inverted pendulum environments ( [cartpole](https://drive.google.com/file/d/19N87JUz4j229XcLrdiVPi85kBJiyp5wl/view?usp=sharing),[dpendulum](https://drive.google.com/file/d/1y_YiHN6xpKuNc0I28Mkl0kozCccp3NUE/view?usp=sharing)). In both cases, the default parameters $\alpha = 0.6$, and $\beta = 0.4$ are shown to be most robust and close to optimal performance on both environments. The original PER paper provided the default hyperparameters after tuning for the Atari environments. We believe that the fact IER outperforms PER on the two Atari environments considered in this work helps further support our case. We also note that being very sensitive to hyperparameters is not good quality for an algorithm. Our proposal is robust (works with p = 0 in most cases) and has only one hyperparameter to tune.
> > > > > >
> > > > > >
> > > > > > As stated in [1], the choice of this hyperparameter ($\beta$) interacts with the choice of prioritization exponent $\alpha$; increasing both simultaneously prioritizes sampling more aggressively at the same time as correcting for it more strongly. These choices are trading off aggressiveness with robustness, but it is easy to revert to a behavior closer to the baseline by reducing $\alpha$ and/or increasing $\beta$. We do notice a few instances of $\alpha = 1$ being the best, but such settings are not advocated by [1] since we have increased aggressiveness of our sampling, and lost out on robustness.
> > > > > >
> > > > > >
> > > > > > [1]  Prioritized Experience Replay (Schaul et al.)
> > > > > >
> > > > > > ---

---

> > > > > > > ### Author Response · Authors · 2022-11-29
> > > > > > > **Gentle Reminder**
> > > > > > >
> > > > > > > We thank you again for your response. We hope we have addressed all your concerns and that you are willing to advocate the paper for acceptance. Please let us know if there are additional concerns we can discuss since the discussion period will end soon. As mentioned in your response, we also hope you bump up the score.

---

> > > > > > > > ### Author Response · Authors · 2022-12-02
> > > > > > > > **Additional results**
> > > > > > > >
> > > > > > > > To expand on our previous results regarding the PER. We perform a robust grid search across all environments we have experimented with. In short, we test with $\alpha$ and $\beta$ in the range $[0.2,0.4,0.6,0.8,1]$. We detail our finding below:
> > > > > > > > - On environments such as [CartPole](https://drive.google.com/file/d/19N87JUz4j229XcLrdiVPi85kBJiyp5wl/view?usp=sharing) , [Reacher](https://drive.google.com/file/d/12XnoACAEeoTRjRui3C2nattXY3itLi6X/view?usp=sharing), we notice only a marginal differences between default hyper parameter which we used and the best hyper parameters.
> > > > > > > > - For other environments such as [HalfCheetah](https://drive.google.com/file/d/1vsTcvd7l6knxfb-aVlx5crePgCLlk7k-/view?usp=share_link), [Ant](https://drive.google.com/file/d/1rAaOZy5aYr3T6OAis9vgW-R6pa5v-ESs/view?usp=share_link), [FetchReach](https://drive.google.com/file/d/18a4Ayl8mITVzRMy1Xv4YbQr55ufmtx3I/view?usp=share_link), [Walker](https://drive.google.com/file/d/1R2geCScXH7dCyhN3FYJz9Sf7ZfHzYVdB/view?usp=share_link), we notice a significant gap between the performance of PER and IER even after a thorough tuning of PER hyperparameters. For instance, IER outperforms PER by more that 8000 in terms of average return on HalfCheetah environment. Furthemore, IER outperforms PER by almost 48 in terms of average return on FetchReach. Finally, IER outperforms PER by more than 6200 in terms of average return on Ant. On Walker, IER outperforms PER by almost 1500 in terms of average return.
> > > > > > > > - Some environments such as [Acrobot](https://drive.google.com/file/d/1FMdUcqhwgxie5sUPXE_LBGY9j1Q0kSmQ/view?usp=share_link), [Pendulum](https://drive.google.com/file/d/1NkAYSiO8M8Ixpc21ZYNviWc3v-sKjqLr/view?usp=share_link), [LunarLander](https://drive.google.com/file/d/1tjqvPjXnhLv4NTQ7mfmoiwoWMUqMbjsW/view?usp=share_link), [Hopper](https://drive.google.com/file/d/19glPd-lmibLcY5zhv3HAahlbAyvVEWzt/view?usp=share_link), and  [Double-Inverted-Pendulum](https://drive.google.com/file/d/1y_YiHN6xpKuNc0I28Mkl0kozCccp3NUE/view?usp=sharing) did show a lot of improvements when tuned for the prioritization exponent $\alpha$, and bias annealing factor $\beta$. We take the best hyperparameters from the grid-search experiment, and re-run the code to compute top-k metrics (i.e, we pick top 3 out of 5 runs). However, from further experiments, we show that even the select hyperparameter is not robust across seeds, and overall performs worse than our proposed approach of IER across all environments: [Acrobot](https://drive.google.com/file/d/1ggrB_oH0A2zvxRWDlLsFXip6GRLBjtIg/view?usp=share_link), [Pendulum](https://drive.google.com/file/d/1bvAWRBuqiNb-KEZgDboOgILTbgMAxJyg/view?usp=share_link), [LunarLander](https://drive.google.com/file/d/1xAiuWZQKoy73RG0ypNuICBOJdgE3XQ3K/view?usp=share_link), [Hopper](https://drive.google.com/file/d/1qNMit7pzwPAhl__28O7Oqoawm2TIMLkj/view?usp=share_link), and [Double-Inverted-Pendulum](https://drive.google.com/file/d/1yfyNAywe_JZF9nnTrmPQzOeiq9fJGNGw/view?usp=share_link).
> > > > > > > >
> > > > > > > > Please check below table for updated comparison between PER and IER below:
> > > > > > > >
> > > > > > > > | _Dataset_ | _PER_ | _IER_ |
> > > > > > > > |:---:|:---:|:---:|
> > > > > > > > | _CartPole_ | 198.06 ± 3.68 | **-199.83** ± 0.31 |
> > > > > > > > | _Acrobot_ | -222.45 ± 121.71 | **-193.90** ± 57.56 |
> > > > > > > > | _Inverted Pendulum_ | -155.69 ± 10.75 | **-150.27** ± 9.63 |
> > > > > > > > | _LunarLander_ | 11.21 ± 15.51 | **12.32** ± 27.55 |
> > > > > > > > | _HalfCheetah_ | 99.75 ± 1124.46 | **10544.88** ± 345.22 |
> > > > > > > > | _Ant_ | -2699.84 ± 1.34 | **4203.21** ± 345.22 |
> > > > > > > > | _Reacher_ | -5.42 ± 0.61 | **-4.92** ± 0.27 |
> > > > > > > > | _Walker_ | 1709.48 ± 1635.90 | **4349.29** ± 680.35 |
> > > > > > > > | _Hopper_ | 784.00 ± 1536.00 | **3205.05** ± 406.35 |
> > > > > > > > | _Inverted Double Pendulum_ | 8654.78 ± 981.67 | **9067.69** ± 402.39 |
> > > > > > > > | _Pong_ | 17.02 ± 3.27 | **19.10** ± 1.20 |
> > > > > > > > | _Enduro_ | 565.23 ± 116.36 | **586.32**  ± 111.44 |
> > > > > > > >
> > > > > > > > We will update the PER numbers to include the results of the grid search.
> > > > > > > >
> > > > > > > > ---
> > > > > > > >
> > > > > > > > We hope this answers the reviewer’s concerns regarding our experiments with PER and you are willing to bump your score to acceptance. Thank you!

---

### Official Review · Reviewer_nF51 · 2022-10-25

**Confidence:** 4
**Clarity, Quality, Novelty And Reproducibility:** The quality and clarity are good. The…
**Correctness:** 3
**Technical Novelty And Significance:** 3
**Empirical Novelty And Significance:** Not applicable
**Recommendation:** 6

**Strength And Weaknesses:**

> Strength

The problem is important and very interesting.

Considering the pivot points and selecting transitions before outcomes seems to be new for experience replay.

The experiments show the proposed method works better than others.

>Weaknesses

The paper argues that the previous methods have bias. Does the proposed have bias too? The paper needs to provide an analysis about it.

The surprised pivots are one of key components of the method. How to pick pivots seems to be not new. The paper uses TD error for selecting pivots. What is the difference between the proposed method and PER about the idea?

Related works are not enough. For example, for the experience replay, the paper does not mention CER (Competitive experience replay), CHER (DHER: Hindsight experience replay for dynamic goals), DHER (Curriculum-guided hindsight experience replay), and so on.

For the important function, it uses the magnitude of the TD error, what does magnitude mean? What exactly is the important function? It is not clear.

It is better to improve the writing of the paper. For example, in Figure 1, the paper uses colors but does not provide explanation.

In Table 2, why does not  IER work better in Pong?

In Table 2 and Table 3, why do IER and Reverse have different results?

The baselines used in the paper are a little confusing. What is OER? Why does it not appear in Table 2? It is better to provide more details.

Why does not the paper provide learning curves for comparing different methods? Using curves is common to present results.


**Summary Of The Paper:**

This paper proposes an experience replay method for reinforcement learning. The paper argued that previous methods are sub-optimal and have bias. The proposed method picks some pivot points at first. Then it selects transitions before these pivot points.

The intuition of the paper is that an agent should select transitions that associate with outcomes.

In the experiments, the proposed method compares UER, PER, HER with multiple environments. The paper uses the top-k seeds moving average return as the evaluation metric and 3 seeds. The paper also compares IER forward and IER reverse. The results show that the proposed method works better for the most of the dataset.


**Summary Of The Review:**

The paper argues the previous methods have bias. However, the paper does not provide enough support to show their method does not have bias. The proposed method uses TD for selecting pivots. It might be similar to previous works. The paper also needs to provide more details about related works and techniques.

---

> ### Author Response · Authors · 2022-11-08
> **Response to the Review**
>
> We thank the reviewer for the detailed comments and questions which have helped us improve our manuscript. We refer to the revised manuscript and the common comments above. We have added three more experiments, and IER outperforms all other baselines in all of them. Please refer below for the responses to specific questions.
>
> ---
>
> - *"The paper argues that the previous methods have bias. Does the proposed have bias too? The paper needs to provide an analysis about it."*
>
> We have discussed this in some detail in Section 6, “Issues ….”. Although it mitigates bias, we acknowledge that our approach is not entirely free from it since we pick pivot points with respect to the TD error. This motivates us to involve a uniform mixing factor with our proposed approach, further reducing spurious correlations. However, we point out that given the superior performance over a diverse set of environments, this bias is necessarily small.  We intend to perform a theoretical analysis of this algorithm in future work, which can allow us to isolate the source and magnitude of the bias.
>
> ---
>
> - *"The surprised pivots are one of key components of the method. How to pick pivots seems to be not new. The paper uses TD error for selecting pivots. What is the difference between the proposed method and PER about the idea?"*
>
> The critical component of our method is not selecting the pivot points but the fact that we want to “look back after picking these pivot points”. PER, as mentioned in Section 2.1, samples experiences from a probability distribution that assigns higher probability to experiences with significant TD error and is shown to boost the convergence speed of the algorithm. However, it needs to sample freshly to obtain each data point, which makes the process computationally inefficient. Our method only requires sampling one pivot point, from which we look backward, making it more efficient. Our approach sets the hyper-parameter p to 0 most of the time and still outperforms other baselines, whereas PER requires a careful hyper-parameter to ensure its proper working.
>
> ---
>
> - *Related works are not enough. For example, for the experience replay, the paper does not mention CER (Competitive experience replay), CHER (DHER: Hindsight experience replay for dynamic goals), DHER (Curriculum-guided hindsight experience replay), and so on.*
>
> Thank you for the references. We have included references to these works in our paper, as well as the references suggested by other reviewers.
>
> ---
>
> - *"For the important function, it uses the magnitude of the TD error, what does magnitude mean? What exactly is the important function? It is not clear."*
>
> TD error is the temporal difference error and is a common term used in reinforcement learning. Magnitude, as the name suggests, refers to us taking the absolute value of the temporal difference instead of focussing on the error sign. Mathematically, the absolute value/magnitude of the TD error is given by $|Q(s,a) - R(s,a) - \gamma \sup_{a^{\prime}}Q^{\mathsf{target}}(s^{\prime},a^{\prime})|$ for state action reward tuple $(s,a,s^{\prime})$. As suggested by the reviewer, we will also clarify this and explain the mathematical computation of TD error in our paper.
>
> ---
>
> - *"It is better to improve the writing of the paper. For example, in Figure 1, the paper uses colors but does not provide explanation."*
>
> We will work to improve the presentation of the paper. The colors were used only to justify that the points comprise a batch, which has also been added explicitly naming each collection as ‘Batch i’. As suggested, we also briefly explain our use of colors in Figure 1 in the modified draft.
>
> ---
>
> - *"In Table 2, why does not IER work better in Pong?"*
>
> Although the mean reward of UER (19.15 ± 1.32) is better than IER (19.10 ± 1.20) in Pong,  the difference is marginal.  It is not entirely clear why UER performs better in these specific cases since the environment itself is very complex. It would be interesting future work to understand the exact environments where IER performs well and where it does not.

---

> > ### Author Response · Authors · 2022-11-08
> > **Response to the Review (Part 2)**
> >
> > - *"In Table 2 and Table 3, why do IER and Reverse have different results?"*
> >
> > As stated in the caption of Table 3, i.e., we use the base setting of IER in this section to avoid spurious comparisons (i.e., with p = 0 and no hindsight). The base setting is crucial to compare apples with apples since the two groups have different properties if we allow hindsight and uniform mixing to influence the training. We stress on this further in the updated draft to avoid any confusion to future readers.
> >
> > ---
> >
> > - *"The baselines used in the paper are a little confusing. What is OER? Why does it not appear in Table 2? It is better to provide more details."*
> >
> > Optimistic Experience Replay (OER) was introduced by  [1] to show that sampling points with the highest TD error come with a bias and sub-optimal performance. This is called “Greedy TD-error Prioritization” in [1]. We have introduced OER early in our work (Introduction as well as Section 3.1), the citation, how it works, and all relevant details. Table 2 shows the readers that our approach outperforms previous SOTA approaches in most of the environments tested. OER being a weaker baseline was not included in Table 2 and instead given its own table with RER in Table 4 for better readability.
> >
> > [1] Prioritized Experience Replay, Schaul et al.
> >
> > ---
> >
> > - *"Why does not the paper provide learning curves for comparing different methods? Using curves is common to present results."*
> >
> > We have provided all the learning curves of our different methods in the Appendix (Appendix C). We have also referenced them from the main text so that our readers do not miss them while reading (Section 5: Comparison with SOTA). Adding the learning curves in the main text would have forced us to shrink them beyond the point of being clearly visible.
> >
> > ---
> >
> > We appreciate your suggestions and are pleased to know that you find the problem not only important and very interesting, but also appreciate the fact that our proposed methodology is novel and outperforms other baselines. Given the fact that we have answered all of the concerns raised by the reviewer,  we hope you are willing to bump our score to acceptance.
> > Please do get back to us if there are any further corrections that could help polish our paper. Again, thank you for your time and effort in reviewing our work!

---

> > > ### Comment · Reviewer_nF51 · 2022-11-30
> > > **Thank you**
> > >
> > > The response addressed most of my questions. However, removing bias might be confusing. I am changing my score.

---

> > > > ### Author Response · Authors · 2022-12-01
> > > > **Response clarifying bias**
> > > >
> > > > Thank you for increasing the score and advocating our paper for acceptance. We further expand on the bias point to avoid confusion. Let us clarify more about the bias. We refer the reviewer to our [response to Reviewer HzYs](https://openreview.net/forum?id=15fiz99C8B&noteId=eGfhqvhEBG) , where we discuss the various meanings of the word bias as used in the literature and in our work and how our method is unbiased with respect to the population gradient in important settings. We hope this helps mitigate some confusion surrounding this topic. Please reach back to us if you have any more questions.

---

> ### Comment · Area_Chair_d9hF · 2022-11-24
> **Thank you! Are you satisfied by the answers?**
>
> Dear reviewer,
>
> Thanks again for your detailed review! The authors have replied back to you. Please read them carefully, and acknowledge their response. If there is still an unclear point about the paper or you do not agree with some of the responses, please let them know. We would like to have a robust discussion now.
>
> If you have any further questions from them, please ask them now. We have to make the final decision soon.
> Also as a courtesy to the authors, please acknowledge their rebuttal.
>
> Thank you,
> Area Chair

---

### Official Review · Reviewer_j7Bo · 2022-10-28

**Confidence:** 3
**Correctness:** 3
**Technical Novelty And Significance:** 3
**Empirical Novelty And Significance:** 3
**Recommendation:** 8

**Clarity, Quality, Novelty And Reproducibility:**

The paper is written clearly and the quality is good. Algorithmic novelty is limited but many insights on experience replay are provided. Code is included in the supplementary material.

**Strength And Weaknesses:**

**Strengths:**
- This paper is well-motivated and a solid contribution to reinforcement learning. The proposed approach is simple and intuitive, and is inspired by recent theoretical groundings of reverse experience replay and combines it with optimistic experience replay.
-  The authors explain the majority of their design choices in detail, compare them with alternative methods, and provide many insights. The example in Section 3.2 and discussion in Section 4 are particularly nice to get a good understanding of different methods. The authors also discuss forward vs. reverse experience replay by noting the prior work of Kowshik et al. 2021 and providing an empirical comparison of the two options.
- The authors report that empirically their approach requires minimal hyperparameter tuning.
- The paper is very well-written.

**Weaknesses:**

I did not find any major weaknesses that would justify rejection. Some details are unclear:
- While the reverse sampling approach is well-justified, the use of TD error for prioritization is not very clear. Why is TD error a good proxy for the surprise pivot? Although Figure 3 shows correlations to rewards, it would be helpful if the authors elaborate more on this. Is prioritizing high TD helpful in breaking the correlations caused by Markovian data collection, and if yes, how? Is it possible that prioritizing high TD samples actually biases the approach to more stochastic transitions, which might not necessarily be a good idea?
- Does the results in Section 3.2 change if the environment is stochastic?


**Summary Of The Paper:**

This paper studies experience replay in reinforcement learning, which is helpful in handling spurious correlations caused by the Markovian data collection process. Prior methods such as uniform and prioritized experience replay can introduce bias and suffer from suboptimal convergence rates. Motivated by recent theoretical advances in experience replay, such as reverse experience replay (RER) (which provably removes bias in the linear setting but is still suboptimal in neural network function approximation), the authors propose introspective experience replay (IER). The IER approach combines ideas from RER, which samples data points in the reverse order, and optimistic experience replay, which samples data greedily according to their corresponding TD error. In summary, the IER approach picks consecutive batches of data before certain "surprising" pivot points, with the highest TD errors. The authors conduct an empirical evaluation of IER on several environments such as classic control, robotics, and Atari, and show that IER outperforms other experience replay methods in most cases.

**Summary Of The Review:**

I found this paper to be a solid contribution to RL. The paper not only introduces a new experience replay algorithm that outperforms prior methods but also provides insight into the limitations of prior methods.

---

> ### Author Response · Authors · 2022-11-08
> **Response to the Review**
>
> We thank the reviewer for the kind comments and insightful questions. We have answered both the concerns raised by the reviewer below. We refer the reviewer to the common comments and the revised manuscript, where we have added three additional experiments. Our method outperforms baselines in all these environments. We are glad that the reviewer finds our work a "solid contribution to RL" and are advocating our paper for acceptance.
>
> ---
>
> - *"While the reverse sampling approach is well-justified, the use of TD error for prioritization is not very clear. Why is TD error a good proxy for the surprise pivot? Although Figure 3 shows correlations to rewards, it would be helpful if the authors elaborate more on this. Is prioritizing high TD helpful in breaking the correlations caused by Markovian data collection, and if yes, how?"*
>
> The explanation is as follows: picking the end-points based on TD error means that we sample states that most likely gave a large positive or negative reward (as shown by correlation to rewards). In most exciting RL environments, high-reward states are sparse and are rarely reached by naive policies used initially. Therefore, IER picks such points preferentially and propagates back to understand how to (or not) reach such high reward ( or high negative reward) states. We do not believe prioritizing high TD helps break the correlations caused by Markov data. This property has only been established for reverse sampling. It would be interesting for future work to study the properties of prioritized sampling in a theoretically rigorous way.
>
> ---
>
> - *"Is it possible that prioritizing high TD samples actually biases the approach to more stochastic transitions, which might not necessarily be a good idea? Does the results in Section 3.2 change if the environment is stochastic?"*
>
> We agree with the possible drawbacks in stochastic environments - here, rare events might keep getting picked and create lots of bias. This is why we avoided considering IER for a long time until we witnessed its superior performance in many popular environments. We modified the example in Section 3.2 by making the transition stochastic with a probability of going left with right action to be 0.2 (and vice versa). However, the trap state seems to be getting picked most of the time, and the goal state is ignored for both IER and OER, with IER still being much better off. However, removing the trap state makes it work as expected.
>
> We are currently working on a similar algorithm that behaves better under stochastic conditions. Here, we pick a 'surprise state' after averaging over all occurrences of the same state-action in the buffer. We then pick a pivot uniformly at random from all the occurances of the 'surprise state'. This leads to fewer 'rare events' biasing the learning process. This is based on the mathematical principle that when we have transitions of the form $(s,a,s^{\prime})$, picking by any importance given by a function of $(s,a)$ only does not bias the Q learning process, but looking at whole tuple $(s,a,s^{\prime})$ to evaluate the importance results in a selection bias. Note that we have averaged out the randomness in $s^{\prime}$ in this new proposal and hence expect this importance metric to be a function of $(s,a)$ only.
>
> ---
>
> Please do get back to us if there are any further corrections that could help polish our paper. Again, thank you for your time and effort in reviewing our work.

---

### Official Review · Reviewer_FAX4 · 2022-11-02

**Confidence:** 5
**Correctness:** 1
**Technical Novelty And Significance:** 1
**Empirical Novelty And Significance:** 1
**Recommendation:** 1

**Clarity, Quality, Novelty And Reproducibility:**

Clarity, quality: bad. Please see "paper organization" above.

Novelty: minor. Prioritizing on TD errors has been extensively studied. Including recent samples too. The novelty would be in the combination of the two, but the one proposed is not sound.

Reproducibility: the description of the benchmarks is only partial but the code is provided (I skimmed through it quickly but did not check in detail nor run it).

**Strength And Weaknesses:**

### Strengths

The paper tries to capture the idea that rewards should be propagated backwards, starting from states which have the largest TD error.

### Weaknesses

The idea that rewards should be propagated backwards in time is the key idea behind asynchronous DP and prioritized sweeping (Moore & Atkeson, 1993) so it is nothing new.
Additionally, the authors miss the fact that Deep RL is about using NNs to solve the Bellman equation, which in turn is a series of empirical risk minimization (approximate dynamic programming) problems over a replay buffer. And ERM suffers from arbitrary sampling schemes.
Overall, there are no strong arguments for the presented method (neither formal nor empirical).

### Algorithmic contribution

The contribution is minor and not really supported by anything else than claims and vague intuitions.
It is not connected to the relevant literature on ER. Especially, there could be connections to well grounded approaches like (Gruslys et al, 2018) or (Zhang & Sutton, 2017).
Gruslys, A., Dabney, W., Azar, M. G., Piot, B., Bellemare, M., and Munos, R. The reactor: A fast and sample-efficient actor-critic agent for reinforcement learning. In International Conference on Learning Representations, 2018.
Zhang, S. and Sutton, R. S. A deeper look at experience replay. In NeurIPS 2017 Deep Reinforcement Learning Symposium, 2017.

The algorithm itself is unclear and not really discussed.
In particular, the way the "importances" are computed is never mentioned. Do you cycle through all the replay buffer at every gradient step to assess these priorities? That sounds very costly.
How do you guarantee all states will eventually be updated?
How does function approximation come into play?
What is the exploration strategy?

### Theoretical soundness

Everything relies on the claim made in Kowshik et al. (2021a and b), which are not recalled. So the soundness of the algorithm is not really backed by prior work.
Besides this, training a neural network remains an empirical risk minimization process, which requires iid samples. The distribution of samples itself can be altered via importance sampling. But making the sampling distribution deterministic is a strange (very doubtable) practice here. Mixing with a uniform distribution might be relevant but is not defended by any sort of analysis.
Additionally, besides the claims that theoretical results are in previous papers, there is no new formal result in this paper.

All the arguments are rather vague and seemingly intuitive.

Lots of repetition about the bias introduced by Markovian data. But Markovian data is never defined. And why it causes bias is never explained (not even intuitively).

### Literature coverage

I'd like to encourage the authors to escape their comfort zone. This work builds upon the contributions by Kowshik et al. (2021a and b) and to some extent Agarwal et al. (2021) which is essentially the same authors on very close topics. Additionally, OER (Optimistic ER) is presented by the authors as introduced in the PER paper, where it is nowhere to be found (and the search in a web browser about optimistic ER returns no relevant link).

This whole work is based on RER, which is introduced in a preprint from 2019 which never received peer-reviewing. RER relies on keeping trajectories in memory and replaying them backwards for updates. I will add, to the best of my ability to check facts, RER was never validated, even experimentally. While keeping the structure of trajectories within the replay buffer might be relevant, there are works out there which go way beyond that.

I fail to understand how one can introduce a comparison with HER. Despite the name, HER is not a non-uniform sampling method for experience replay. It is an intrinsic motivation method for designing goal-based policies. So the point of the comparison evades me.

Most of the abundant literature on ER is missing in this paper. To the best of my knowledge, the most recent paper on the topic is by Lahire et al. (2022), which contains a quite extensive literature survey about other recent work in ER. This same paper establishes a link between selecting specific samples in a replay buffer for SGD-based Bellman updates, and importance sampling to reduce the variance of SGD's gradient estimate. I encourage the authors to check this work and the corresponding body of literature.
Lahire, T., Geist, M., & Rachelson, E. (2022). Large Batch Experience Replay. International Conference on Machine Learning.

### Empirical evaluation

The toy example is very incomplete. Is there a function approximator in the form of a NN? If yes, what is the NN's input? The state index? Is there a learning rate decrease schedule?
I think this toy example is very biased and the only thing it actually shows (in a vague fashion) is that *maybe*, in *some* function approximation cases, asynchronous DP updates may possibly lead to an optimal value function regardless of the samples distribution for SGD updates.

The "broad category of environments" claimed is actually a set of 10 environments, including some very simple ones (cartpole) and only 2 ALE environments (Pong and Enduro, why only these two?). This seems very light. Why this choice?
The top-k seeds is a very questionable choice of performance metric. Please avoid doing this. It uselessly biases results. We don't need RL agents that perform well once in a while.
I strongly doubt some claims, such as the speedup. With only a partially defined algorithm, how can I evaluate this? What is the baseline for comparison? what are the actual figures, rather than just reporting percentages of improvement?

Overall the experimental validation is unconvincing.

### Paper organization.

Very bad organization and writing overall:
- Constant forward references, to section 4 from section 2, to section 5 from table 1 (which should actually be in section 5, not at the beginning of the paper).
- Lots of references to the appendix, for important results, backing claims from the main text (and nothing in the main text about the same claims).
- Excessive use of superlatives for unsupported claims (minimal tuning, important metric...).
- Blur, unsupported statements ("a broad category of environments" with no details, "a few Atari environments",... "many other classes of environments", "a total of 1e6 data points usually").
- Lots of typos (e.g. "important" instead of "importance", "staring" instead of "starting")
- Misinterpretation of the assignment operator "$\leftarrow$".
- Poor English.
- Repetitions about what is in the appendix and too little discussion in the main text.

**Summary Of The Paper:**

This paper proposes to sample experience from a replay buffer by keeping a memory of trajectories, and picking sequences of B consecutive samples as minibatches. The sequence is the one whose last state features the largest TD error. Each sequence can be mixed with samples drawn uniformly from the replay buffer.

**Summary Of The Review:**

This paper proposes a sampling scheme for emphasizing states with large TD error and their B predecessors along trajectories, when computing gradients for SGD updates in approximate value iteration algorithms. The algorithm's presentation is very incomplete, the ideas are poorly motivated, the empirical evaluation does not permit drawing conclusions and the paper's organization can be much improved. I recommend rejection.

---

> ### Author Response · Authors · 2022-11-08
> **A Response to the Review.**
>
> We thank the reviewer for the feedback, especially regarding the additional references, which we have included in the modified manuscript. We have also added three new experiments - pendulum, walker, and reacher as discussed in the common comments.
>
> We apologize for the lengthy response. However, this is necessitated by the fact that the review contains a number of factually incorrect claims and claims based on conceptual misunderstandings of RL. Most of the missing details and weaknesses quoted by the reviewer have already been extensively discussed in the paper. For the sake of clarity, we divide such comments from the reviewer into the following categories and classify them.
>
>
> **Omissions:**
>
> **a)** One of the paper's main conclusions, backed by multiple experiments and extended discussions (but ignored).
>
> **b)** Important details of the Algorithm which has been mentioned clearly in the main body (but ignored).
>
> **c)** Discussions for which we have dedicated an entire paragraph in the main text (but ignored).
>
> **d)** Standard details of the RL algorithms which have been explicitly given in the appendix, and are common across all benchmark (but ignored)
>
> **Fallacies:**
>
> **e)** Criticism and observations which are irrelevant to the workings of Experience replay and/or are incorrect from the conceptual viewpoint of RL.
>
> **f)** Criticisms based on logical fallacies and arbitrary dismissals without explanation.
>
> (comments by the reviewer in italics)
>
> ---
>
> - *“The idea that rewards should be propagated backwards in time is the key idea behind asynchronous DP and prioritized sweeping (Moore & Atkeson, 1993) so it is nothing new”*
>
> We have not introduced it as a new observation and have cited this work in Section 2.2. Several cited works in this section make this connection. Note that DP methods, in general, and Moore-Atkeson's method, in particular, are planning algorithms that assume that the model is known (i.e., transition probabilities and rewards are known or estimated beforehand). This is not the case with online, model-free RL, which is unaware of the model and has to explore. Although relevant, the connection with prioritized sweeping is not the whole story.
>
> ---
>
> - *“Additionally, the authors miss the fact that Deep RL is about using NNs to solve the Bellman equation, which in turn is a series of empirical risk minimization (approximate dynamic programming) problems over a replay buffer. And ERM suffers from arbitrary sampling schemes.”*
>
>  This is factually incorrect and falls in **category (a)** above. We acknowledge that ERM suffers from arbitrary sampling schemes in our OER and IER(F) discussion. In fact, in both the toy example and the ablation studies, we show that such naive sampling schemes do not work. **One of the critical messages of the paper is that our sampling scheme (IER) does not seem to suffer from these drawbacks, which has been reiterated multiple times.**
>
> ---
>
> - *“In particular, the way the "importances" are computed is never mentioned.”*
>
> This is not true and falls in **category (b)** above. We have mentioned this prominently in Algorithm 1 (the main algorithm). Refer to the line with the comment, “Compute the importance of each data point in the buffer.”
>
> ---
>
> - *“Do you cycle through all the replay buffers at every gradient step to assess these priorities? That sounds very costly.”*
>
>  Yes, as stated above. This is also used in prioritized experience replay and topological experience replay. We have dedicated an entire paragraph to explain this in Section 6. Here we propose considering `lazy updates’ for speedups and relegating this to future work. This falls in **category (c)** above.
>
> ---
>
> - *“How do you guarantee all states will eventually be updated?”*
>
> In the environments considered in the paper, the state space is vast, and all states cannot be updated as required by tabular RL guarantees. In structured state spaces, it is not necessary to visit all the states either. This is the main reason function approximation is used (see Barto and Sutton’s book for details). In terms of modern RL algorithms that use neural approximation, exploration guarantees are primarily unavailable, and the performance against the benchmarks is taken as the evaluation standard. This comment falls in **category (e)** above since the premise is incorrect from a primary conceptual view point.

---

> > ### Author Response · Authors · 2022-11-08
> > **Continuation of Our Response (Part 2)**
> >
> > - *“How does function approximation come into play? What is the exploration strategy”*
> >
> > Function approximation is irrelevant to the description of ER. ER which can be deployed with any function approximation (falls into **category (e)** above). However, we note that the empirical results are given with neural function approximation (as described in the paper). The exploration strategy, too, is a part of the RL algorithm and not relevant to the ER method. We use the standard epsilon greedy strategy - and have described the parameters in the appendix ( **category (d)** above). It is standard in RL papers to relegate such details to the appendix.
> >
> > ---
> >
> > - *“Everything relies on the claim made in Kowshik et al. (2021a and b), which are not recalled. So the soundness of the algorithm is not really backed by prior work.”*
> >
> > Describing these works' results requires extensive notation and would cause a significant digression from the current work. We will try to fit in more details as appropriate. It is non-sequitur to claim that the soundness of the algorithm is not backed by prior work, given that we have cited the prior work and these works are publicly available. This falls into **category (f)**.
> >
> > ---
> >
> > - *“Besides this, training a neural network remains an empirical risk minimization process, which requires iid samples. The distribution of samples itself can be altered via importance sampling. But making the sampling distribution deterministic is a strange (very doubtable) practice here.”*
> >
> > This is incorrect from a fundamental learning theory perspective. Empirical risk minimization can be done with any data, but i.i.d. data seems to work well in practice. It does not mean it cannot work well with other sorts of data. Several works have shown that empirical risk minimization can be efficient with non-i.i.d data as well, with or without ergodicity (see [1a],[1b],[1c] and references therein). Even in RL, fitted Q iteration does ERM with respect to bellman error but with non-iid data. It has been noted in the literature that the use of a target network makes DQN closer to fitted Q iteration (see [2b])
> >
> > The main problem in RL is not with ERM, but with the fact that we use mini-batch SGD type algorithms to optimize. Here, using correlated batches (as in naive forward pass) do not converge to the ERM solution. (See the extensive experiments in Kowshik et al 2021 a, where ERM is compared to forward SGD and SGD RER).
> >
> >  We refer to the reverse experience replay-based theoretical works (Agarwal et al., 2021, etc.), which show that going in the reverse direction with SGD brings out the super martingale structure in error, and this makes the data points behave like i.i.d data, from the perspective of concentration inequalities. This comment falls into **category (e)**.
> >
> >  Even though the sampling procedure is deterministic, there is still randomness due to the epsilon greedy procedure used to gather the data and the randomness in the transitions of the MDP itself. The reviewer dismisses the method as strange and doubtable. However, we refer to the experimental results, which demonstrate the algorithm's superior performance. **This comment by the reviewer is non-sequitur unless the reviewer calls into question our experiments' veracity**.
> >
> > [1a] Learning Without Mixing, Simchowitz et al
> >
> > [1b] Non-asymptotic and accurate learning of nonlinear dynamical systems, Oymak and Sattar
> >
> > [1c] Learning with little mixing, Ziemann and Tu
> >
> > ---
> >
> > - *“Mixing with a uniform distribution might be relevant but is not defended by any sort of analysis”*
> >
> > We have already cited and discussed two papers (Hong et al (2022) and Lee et al (2019)) which utilize mixing with other forms of ER with great results (**category (b)**). Regarding theoretical analysis, many works (including theoretical ones) assume that even purely uniform experience replay ([2a] and [2b]) with a large buffer gives i.i.d data, which remains unsubstantiated so far in the literature in a theoretically rigorous way. It is not easy to theoretically analyze such experience replay techniques due to the complexity of the processes involved. Therefore, such an analysis is beyond the scope of this work. This falls into **category (f)**.
> >
> > [2a] A new convergent variant of Q-learning with linear function approximation, Carvalho et al 2020
> >
> > [2b] A Theoretical Analysis of Deep Q-Learning, Fan et al, 2020
> >
> > ---
> >
> > - *“All the arguments are rather vague and seemingly intuitive”*
> >
> > We respectfully disagree with this statement. The didactic toy example demonstrates the behavior we expect intuitively (causality + sparse reward propagation). We also demonstrate that High TD error samples usually have large rewards even in complex environments and show via ablation studies that modifications that violate these explanations do not perform well.

---

> > > ### Author Response · Authors · 2022-11-08
> > > **Continuation of our Response (Part 3)**
> > >
> > > - *"Lots of repetition about the bias introduced by Markovian data. But Markovian data is never defined. And why it causes bias is never explained (not even intuitively)"*
> > >
> > > We have cited theoretical works for details (ex: Nagaraj et al. 2020) - and this is the premise of these cited papers. On an intuitive level, this is because the dynamics of the Markov chain get coupled to the dynamics of SGD/ Q-learning type descent algorithms. **This explanation is the first line of the introduction** (and hence falls in **category (c)**). Space permitting, we will include this explanation in more detail while re-organizing the manuscript. We believe the term `Markovian data' is clear, given that the subject matter of the manuscript deals with reinforcement learning.
> > >
> > > ---
> > >
> > > - *“Additionally, OER (Optimistic ER) is presented by the authors as introduced in the PER paper, where it is nowhere to be found (and the search in a web browser about optimistic ER returns no relevant link)”*
> > >
> > > OER is our terminology since the original PER paper ([3]) calls it “greedy TD-error prioritization,” which did not fit well with our vocabulary. We have clarified this aspect. We also note that OER is obtained by setting $\alpha \to \infty$  in their algorithm.
> > >
> > > [3]  Prioritized Experience Replay, Schaul et al.
> > >
> > > ---
> > >
> > > - *“This whole work is based on RER, which is introduced in a preprint from 2019 which never received peer-reviewing. RER relies on keeping trajectories in memory and replaying them backwards for updates. I will add, to the best of my ability to check facts, RER was never validated, even experimentally.”*
> > >
> > >  This is incorrect. Our “whole work” is not based on the preprint from 2019, as IER is not the same as RER. We make our arguments and inferences based on our experiments (which include RER as an ablation study conducted for every environment we have considered). Therefore, the publication status of the original preprint is irrelevant. This falls into **category (f)**.
> > >
> > > ---
> > >
> > > - *“I fail to understand how one can introduce a comparison with HER. Despite the name, HER is not a non-uniform sampling method for experience replay. It is an intrinsic motivation method for designing goal-based policies. So the point of the comparison evades me.”*
> > >
> > > HER is a popular experience replay method that is extensively used in practice. Hence is a valid benchmark for comparison.
> > >
> > > ---
> > >
> > > - *"The toy example is very incomplete. Is there a function approximator in the form of a NN? If yes, what is the NN's input? The state index? Is there a learning rate decrease schedule?"*
> > >
> > > This is a tabular MDP, and we have not used NNs here. We used a constant step size and have updated the paper to include these details in the appendix. This example has very few states, so function approximation is unnecessary.
> > >
> > > ---
> > >
> > > - *“I think this toy example is very biased and the only thing it actually shows (in a vague fashion) is that maybe, in some function approximation cases, asynchronous DP updates may possibly lead to an optimal value function regardless of the samples distribution for SGD updates. "*
> > >
> > >  We want to emphasize that toy examples - by their nature - are simple examples illustrating the workings of complex methods on a basic, intuitive level. We believe our model succeeds in this endeavor. ‘Toy examples’, as the name suggests, have been used similarly in all fields of science.  This is why we consider a one-dimensional example with tabular Q-learning. This comment falls into **category (f)**.
> > >
> > > ---
> > >
> > > - *“The "broad category of environments" claimed is actually a set of 10 environments, including some very simple ones (cartpole) and only 2 ALE environments (Pong and Enduro, why only these two?). This seems very light. Why this choice?”*
> > >
> > >  Thanks for pointing this out. We have evaluated three more environments and have included them in the paper, bringing the total to 13 (see the common comment for details). We only included two atari games since we wanted to demonstrate that the method performs well with minimal hyperparameter tuning across a diverse range of environments (only the parameter $p$, which too is $0$ most of the time). It is common for RL works to evaluate on less than ten environments, and we cite many highly regarded works which do this, with a few of them being Atari Games ([4a],[4b],[4c],[4d]).
> > >
> > > [4a] Decision Transformer: Reinforcement Learning via Sequence Modeling, Chen et al
> > >
> > > [4b] Soft Actor-Critic: Off-Policy Maximum Entropy Deep Reinforcement Learning with a Stochastic Actor, Haarnoja et al
> > >
> > > [4c] High-Dimensional Continuous Control Using Generalized Advantage Estimation, Schulman et al
> > >
> > > [4d] Continuous Deep Q-Learning with Model-based Acceleration, Gu et al

---

> > > > ### Author Response · Authors · 2022-11-08
> > > > **Continuation of Our Response (Part 4)**
> > > >
> > > > - *“The top-k seeds is a very questionable choice of performance metric. Please avoid doing this”*
> > > >
> > > >  Top k seeds have been used routinely in the literature [5a,5b,5c,5d,5e], which is why we use them. This has also been explicitly stated in the paper (see Section 5 Metrics)
> > > >
> > > > [5a]  Prioritized experience replay. Schaul et al
> > > >
> > > > [5b] Jointly learning to construct and control agents using deep reinforcement learning. Schaff et al
> > > >
> > > > [5c] Rl-gan-net: A reinforcement learning agent controlled gan network for real-time point cloud shape completion. Sarmad et al
> > > >
> > > > [5d] Scalable trust-region method for deep reinforcement learning using kronecker-factored approximation. Wu et al
> > > >
> > > > [5e] Asynchronous methods for deep reinforcement learning. Mnih et al
> > > >
> > > > ---
> > > >
> > > > - *“It uselessly biases results. We don't need RL agents that perform well once in a while.”*
> > > >
> > > > This is not entirely correct. Consider the train-and-deploy situation where the RL agent is trained in a benign environment and then deployed in a real-world environment where it is dangerous to continue exploring new policies. Here, the agent must learn a good policy once (after many tries). This is why top k seeds are considered a relevant metric in RL. This falls in **category (f)**.
> > > >
> > > > ---
> > > >
> > > > - *"Overall the experimental validation is unconvincing."*
> > > >
> > > > We disagree with this comment, and cite all the responses we have provided above.
> > > >
> > > > ---
> > > >
> > > > **Regarding Organization**: We thank the reviewer for the feedback. We will improve the overall organization of the paper based on this.

---

> ### Comment · Area_Chair_d9hF · 2022-11-24
> **Thank you! Are you satisfied by the answers?**
>
> Dear reviewer,
>
> Thanks again for your detailed review! The authors have replied back to you. Please read them carefully, and acknowledge their response. If there is still an unclear point about the paper or you do not agree with some of the responses, please let them know. We would like to have a robust discussion now.
>
> If you have any further questions from them, please ask them now. We have to make the final decision soon.
> Also as a courtesy to the authors, please acknowledge their rebuttal.
>
> Thank you,
> Area Chair

---

### Author Response · Authors · 2022-11-08
**General Comment to All Reviewers and AC**

We thank all the reviewers for their constructive feedback. We will improve the paper's organization, as pointed out by the reviewers. While we work on the revision of the organization, the current version contains our responses to all the technical questions and includes the additional references stated by the reviewers.

We have responded to the technical questions in the individual comments. In response to some concerns about the lack of an adequate number of experiments, we were able to run three more experiments, as shown below. Here too, IER outperformed all other benchmarks in each of these environments. The results from the Table have been updated for your reference.


| Dataset | UER | PER | HER | OER | RER | IER |
|:---:|:---:|---|---|---|---|---|
| Inverted Pendulum | -161.933 ± 10.551 | 171.731 ± 10.55 | -629.42 ± 815.235 | -735.379 ± 613.5 | -166.567 ± 17.233 | **-150.271 ± 9.627** |
| Reacher | -4.97 ± 0.31 | -5.42 ± 0.61 | -5.30 ± 0.50 | -5.91 ± 0.37 | -5.28 ± 0.58 | **-4.92 ± 0.27** |
| Walker | 3597.03 ± 1203.79 | 1709.48 ± 1635.90 | 889.82 ± 1427.92 | -1578.33 ± 1313.11 | 207.51 ± 193.06 | **4349.29 ± 680.35** |

---

### Author Response · Authors · 2022-11-11
**Regarding our response**

We thank the reviewers for their thoughtful feedback! We are encouraged they find the problem important and very interesting (Reviewer nF51 and Reviewer HzYs), well-motivated, and a solid contribution to reinforcement learning (Reviewer j7Bo); Besides, we are glad that reviewers find the approach is supported by ample experiments (Reviewer HzYs, Reviewer nF51, and Reviewer j7Bo). We are also encouraged that they agree the paper is clear, easy to follow, well-written, and reader-friendly (Reviewer j7Bo, Reviewer nF51, and Reviewer HzYs) and explains most of their design choices (Reviewer j7Bo).

We have:
- Answered reviewers' specific comments in our responses and resolved most of the technical concerns.
- Added three additional experiments where our method outperforms all the baselines. This further strengthens our approach.
-  Conducted multiple additional ablation studies to study the effect of batch size, the importance of the TD metric, and the design choice of looking backward from our experiments on the CartPole environment (as suggested by Reviewer HzYs).
- Improved the clarity of our presentation based on the suggestions.


Since the paper revision period closes soon, we hope the reviewers can acknowledge and respond to our response. This will help us make any further improvements.

---

> ### Comment · Area_Chair_d9hF · 2022-11-12
> **Sent a reminder**
>
> Hi,
>
> I just sent a reminder to all reviewers. Hopefully they will engage with you in the next few days.
>
> Thank you,
> Area Chair

---

### Decision · Program_Chairs · 2023-01-20

**Decision:**

Reject

**Justification For Why Not Higher Score:**

The empirical results are questionable. The writing is imprecise.

**Justification For Why Not Lower Score:**

N/A

**Metareview: Summary, Strengths And Weaknesses:**

The paper describes a novel experience replay method. Briefly, the TD error defines pivot points in the experience replay buffer. The method then uses those pivot points and select samples by moving backwards from them. This part is like reverse experience replay. Intuitively, the suggested method looks back when surprised.

The method does not theoretically analyze its properties. The justifications are based on empirical studies, on both a toy problem as well as many standard benchmarks.

Most reviewers believe that the idea is novel, the design choices are described well, and the empirical results show the superiority of the method.

On the negative side, the paper is a bit imprecise. For example, the paper mentions that previous methods have bias as one of their main weaknesses but does not actually show that their method is unbiased. In fact, the method leads to a biased sampling of the state space (this is mentioned in Discussion (Section 6) by saying that "Picking pivot points by looking at the TD error might cause more biases"). The fact that the method performs well on some benchmarks is not an indication of its unbiasedness.

Similarly problematic is the use of the terms optimal or sub-optimal, which are defined and used in non-standard ways.

I believe these can be fixed by a major revision in the paper.

Another major issue with the paper is the use of top-k seeds as the performance metric with the choice of k = 3 (this is mentioned by Reviewer FAX4), which is selected from the total number of seeds n = 5 (not specified in the paper but clarified by the authors during the discussion period).

First, computing the average performance based on 3 results leads to inaccurate estimates of the performance.

Moreover, the top-k seeds, even though used previously by some papers, is a problematic metric. To see why, consider random variables $X_1, X_2, ..., X_n$ all distributed according to a Gaussian distribution. Suppose $X_i$ is the performance of one of the seeds. The expected value of $\max_i X_i$, which corresponds to the choice of top-1, is going to behave as $O(\sqrt{\log(n)})$. That is, as you increase n, the top-1 seed performance increases in an unbounded way. So if one chooses n to be a large number, but just report the top-3, the performance could be overestimated greatly.
This is not limited to a Gaussian distribution though. The top k out of n for any distribution is biased upwards. How much bias depends on the distribution and the choice of k and n.

The unsuitability of top-k seeds metric is discussed in Henderson et al., "Deep reinforcement learning that matters," AAAI, 2018, which is already cited by the paper, as well as Agarwal et al., "Deep Reinforcement Learning at the Edge of the Statistical Precipice," NeurIPS, 2021, which is not cited. Look at Section "Random Seeds and Trials" of the former and Appendix A.4 of the latter.

I strongly suggest that the authors re-run their experiments with much larger number of seeds/runs (perhaps in the order of 20-30 for most experiments, perhaps except for Atari ones) and report the average and standard error. If the authors insist on reporting the top-k performance as well, that is OK, but they need to explain what it is measuring. It is not measuring the average performance of the algorithm. Perhaps it is an estimate for the k/n-th quantile. This should be clarified.

Overall, I believe this paper has some room for improvement, both in the quality of reported empirical results and the clarify and preciseness of its writing. Therefore, unfortunately I cannot recommend its acceptance.